

# A self-consistent, multi-variate method for the determination of gas phase rate coefficients, applied to reactions of atmospheric VOCs and the hydroxyl radical

Jacob T. Shaw[1], Richard T. Lidster[1*], Danny R. Cryer[2], Noelia Ramirez[2], Graham A. Boustead[2], Lisa K. Whalley[2,3], Trevor Ingham[2,3], Andrew R. Rickard[1,4], Rachel E. Dunmore[1], Dwayne E. Heard[2,3], Ally C. Lewis[1,4], Lucy J. Carpenter[1], Jacqui F. Hamilton[1], Terry J. Dillon[1]

[1]Wolfson Atmospheric Chemistry Laboratories, Department of Chemistry, University of York, Heslington, York, YO10 5DD, UK
[2]School of Chemistry, University of Leeds, Leeds, LS2 9JT, UK
[3]National Centre for Atmospheric Science, University of Leeds, Leeds, LS2 9JT, UK
[4]National Centre for Atmospheric Science, University of York, Heslington, York, YO10 5DD, UK
[*]now at DSTL, Porton Down, Salisbury, Wiltshire, SP4 0JQ, UK

*Correspondence to*: Jacqui F. Hamilton (jacqui.hamilton@york.ac.uk)



**Abstract.** Gas-phase rate coefficients are fundamental to understanding atmospheric chemistry, yet experimental data are not available for the oxidation reactions of many of the thousands of volatile organic compounds (VOCs) observed in the troposphere. Here a new experimental method is reported for the simultaneous study of reactions between multiple different VOCs and OH, the most important daytime atmospheric radical oxidant. This technique is based upon established relative rate concepts but has the advantage of a much higher throughput of target VOCs. By evaluating multiple VOCs in each experiment, and through measurement of the depletion in each VOC after reaction with OH, the OH + VOC reaction rate coefficients can be derived. Results from experiments conducted under controlled laboratory conditions were in good agreement with the available literature for the reaction of nineteen VOCs, prepared in synthetic gas mixtures, with OH. This approach was used to determine a rate coefficient for the reaction of OH with 2,3-dimethylpent-1-ene for the first time; $k = 5.7 \ (\pm 0.3) \times 10^{-11} \ cm^3$ molecule$^{-1}$ s$^{-1}$. In addition, a further seven VOCs had only two, or fewer, individual OH rate coefficient measurements available in the literature. The results from this work were in good agreement with those measurements. A similar dataset, at an elevated temperature of 323 ($\pm$ 10) K, was used to determine new OH rate coefficients for twelve aromatic, five alkane, five alkene and three monoterpene VOC + OH reactions. In OH relative reactivity experiments that used ambient air at the University of York, a large number of different VOCs were observed, of which 23 were positively identified. 19 OH rate coefficients were derived from these ambient air samples, including ten reactions for which data was previously unavailable at the elevated reaction temperature of $T = 323 \ (\pm 10)$ K.





# 1 Introduction

The atmosphere is an extremely complex and reactive mixture containing large numbers of inorganic and organic chemicals (Lewis et al., 2000; Goldstein and Galbally, 2007). Annually, over 1000 Tg of VOCs are emitted to the troposphere from both anthropogenic and biogenic sources (Guenther et al., 2002). They have wide-ranging impacts on human health, either directly

on inhalation, or through their photochemical degradation cycles, which generate polluting and radiatively active secondary products such as ozone and aerosols. Atmospheric oxidative removal of most VOCs is mainly initiated by gas-phase reaction with the hydroxyl radical (OH). This process determines the tropospheric lifetime of an organic compound and often represents the rate-determining step in tropospheric ozone production and photochemical smog formation (Finalyson-Pitts and Pitts Jr., 1997). Photochemical models used for air quality forecasting and future climate predictions can only attempt to represent these

processes if provided with accurate rate coefficients for OH + VOC reactions. Thus, this aspect of gas-phase kinetics represents a fundamental pillar of atmospheric and, more widely, environmental science.

Many hundreds of reactions for the OH initiated degradation of VOCs have been measured in the laboratory. However, many thousands more VOCs are emitted, or are formed in atmospheric photochemical oxidation, for which no OH kinetics have been measured. In order to bridge the gap between our limited fundamental experimental knowledge and the

detailed chemical mechanisms required to describe the chemical schemes employed in air quality and climate models, a number of methods have been developed to estimate rate coefficients for unmeasured species, including OH oxidation (Atkinson, 1986; Kwok and Atkinson, 1995; Ziemann and Atkinson, 2012). A structure-activity relationship (SAR) relates parameters such as rate coefficient to the structural properties of chemical species, thereby providing a method of parameter estimation which does not rely on experimentation. SARs are developed from datasets of experimentally derived parameters, and therefore

require accurate and reliable kinetic information from a wide range of organic reactions (in terms of both structure and functionality) to derive the various group rate coefficients and substituent factors needed (Calvert et al., 2002).

Almost all experimental OH + VOC kinetic data used for atmospheric science have been determined via measurements conducted in controlled laboratory environments. Absolute methods for determining rate coefficients, such as flash photolysis and discharge flow, rely on accurate observations of the VOC concentration, which is maintained in large

excess over that of OH (Pilling and Seakins, 1995). Much of the difficulty associated with these methods is related to the direct monitoring of the short-lived OH radical. Alternatively, the relative-rate method requires neither an accurate knowledge of VOC concentration, nor direct monitoring of OH radicals (Atkinson, 1986). This technique instead relies on the simultaneous measurement of two species: a target VOC that is the focus of the investigation, and a well characterised reference compound, which ideally react at a similar rate. Upon exposure to OH, these two VOCs are depleted relative to their individual rate

coefficients. Thus, the ratio of their depletion allows for the calculation of the target rate coefficient, providing the OH + reference VOC rate coefficient is accurately known and the losses of both compounds in the reactive system are solely governed by reactions with OH. Traditionally, both absolute and relative-rate methods have been used to obtain rate



coefficients for one OH + VOC reaction at a time; as a result, these methods are time consuming and few laboratories are capable of sustaining this type of fundamental science.

Whilst measured rate coefficients for reactions of OH with simple short-chain VOCs (up to seven carbon atoms) have been measured and evaluated at least once, and some many times over, data for many larger, more complex and multifunctional

VOCs are often poorly constrained or unmeasured. Recent observations in a megacity demonstrate that VOCs containing more than seven carbon atoms can make up greater than 50% of the local atmospheric hydrocarbon mass and dominate secondary organic aerosol (SOA) production (Dunmore et al., 2015). As the number of carbon atoms increases, so does the relative complexity of the oxidation reaction scheme, and hence the photochemical potential to form secondary pollutants such as ozone and SOA. As we discover more concerning the complexity of atmospheric VOCs chemistry, the database for OH +

VOC rate coefficients appears increasingly deficient. Conventional laboratory methods, limited to studying a single VOC at a time, are no longer adequate and as such new approaches and techniques are required.

Comprehensive measurements of each individual VOC + OH reaction rate coefficient can be avoided by using direct OH reactivity measurement techniques to measure the total OH reactivity sink. This parameter is equivalent to the inverse of the lifetime of the OH radical and can vary greatly depending on total VOC loading, from milliseconds in heavily polluted

areas, to tens of seconds in clean air (Yang et al., 2016). Techniques using direct laser induced fluorescence (LIF) detection of OH first identified a significant mismatch between measured and modelled OH reactivities (Kovacs et al., 2003; Di Carlo et al., 2004, Edwards et al., 2013). More recently, a technique that does not require the direct observation of OH radicals, termed the comparative reactivity method (CRM), has been developed (Sinha et al., 2008). It is thought that a lack of kinetic data for many OH + VOC reactions may contribute to the so-called 'missing reactivity' that has been observed both locally and globally

(Lidster et al., 2014).

We describe here an experimental technique capable of measuring rate coefficients for large numbers of OH + VOC reactions simultaneously, which builds upon a method employed by Kato et al. (2011) to measure the contribution of unidentified VOCs to OH reactivity. Coupling a conceptually simple and internally self-consistent relative rate kinetic method to modern automated analytical equipment allows for the study of a wide range of species, including some that are difficult to

measure using more conventional methods. Rather than rely on one reference VOC, a subset of the existing OH + VOC kinetic database is used to place the experimental relative rate data on an absolute scale, thereby reducing uncertainties. A broad range of OH + VOC reactions were studied, comparisons with the literature made and rate coefficients for previously unmeasured OH + VOC reaction determined at two reaction temperatures.

## 2 Methodology

Experiments were conducted under near ambient conditions ($P = 1.0$ bar ($N_2$), $T = 295\text{-}323$ K) in a stainless steel flow reactor (approximate external dimensions $470 \times 25 \times 25$ mm, internal volume 250 cm$^3$). A schematic of the apparatus is shown in Fig. 1. A quartz window, positioned on the top face of the reactor, allowed collimated vacuum-ultraviolet (VUV) light from a low-



pressure Hg/Ar lamp (L.O.T., Pen-Ray®) to enter the tube. $N_2$ (2000 standard cubic centimetres min$^{-1}$ (sccm)) was passed through a bubbler filled with high purity water (Fischer, Optima grade) and supplied to the reactor upstream of the quartz window via a mass flow controller (MKS). OH was generated in the reaction chamber via the photolysis of $H_2O$ via R1. Atomic H is rapidly converted to $HO_2$ via R2. The oxygen is found as impurities in the $N_2$ carrier gas.

$$H_2O + h\nu \text{ (184.95 nm)} \rightarrow OH + H \tag{R1}$$
$$H + O_2 \rightarrow HO_2 \tag{R2}$$

The VOC mixture was introduced downstream of the quartz window via a stainless-steel injector (external diameter 3.2 mm)
to avoid unwanted removal of organics via photolysis. A sliding injector allowed for the optimisation of the inlet location with respect to both minimising the effects of VUV photolysis and maximising exposure to the short-lived OH radicals (Cryer, 2016). Mixing of the VOC flow (200-1000 sccm) with the main OH / $H_2O$ / $N_2$ flow was aided by the injector design, which ensured that VOCs pass into the main body of the reactor via an array of four holes radially distributed about the injection tube. The VOC mixture, along with a secondary flow of $N_2$, were controlled by two mass flow controllers (Tylan), allowing
for the variation of VOC mixing ratio in the reactor. This meant that testing samples with a range in total VOC loading and OH reactivity was possible. A brief outline of the estimated range in total VOC concentrations and OH reactivity of each mixture injected into the reactor can be found in Supplementary Information Table S1. Assuming a constant total flow rate of 3000 sccm the residence time of the VOCs in the reactor was calculated to be approximately 4 s, of which OH is estimated to persist for no longer than 0.5 s.

By alternating between the lamp switched off and the lamp switched on, a set of observations was generated that could be split into two unique datasets; those with OH initiated depletion and those without. The depletion of an individual VOC, with a known OH rate coefficient can be evaluated using simple kinetic equations. This is shown using isoprene as an example in Eqs. (1), (2) and (3). The depletion in a VOC due to reaction with OH ($\ln\left(\frac{[VOC]_0}{[VOC]}\right)$) can be related to its rate coefficient, $k$, as shown in the example in Eq. (3).

$$\text{isoprene} + OH \xrightarrow{k_2} \text{(products)} \tag{R3}$$
$$\frac{d[\text{isoprene}]}{dt} = -k_2[\text{isoprene}][OH] \tag{1}$$
$$\frac{d[\text{isoprene}]}{[\text{isoprene}]} = -k_2[OH]dt \tag{2}$$
$$\ln\left(\frac{[\text{isoprene}]_0}{[\text{isoprene}]_t}\right) = k_2 \int [OH]_t dt \tag{3}$$

The integral of OH concentration over time ($\int [OH]_t dt$) is known as the OH exposure ($OH_{exp}$). The assumption is that all VOCs in the sample experience identical exposure to OH radicals owing to rapid homogenous mixing in the reactor. However, despite



the establishment of laminar flow in the reactor, thorough and complete mixing is unlikely to take place. When using reactive gas mixtures containing VOCs which react rapidly with OH, a non-linear relationship between $\ln\left(\frac{[\text{VOC}]_0}{[\text{VOC}]}\right)$ and $k$ may be observed, as compounds experience differing exposures to OH. Slow reacting VOCs may experience a larger exposure to OH relative to faster reacting VOCs. However, this does not represent a limitation to the technique. So long as a relationship

between $\ln\left(\frac{[\text{VOC}]_0}{[\text{VOC}]}\right)$ and $k$ can be characterised over a range of rate coefficients, and the data placed on an absolute scale, relative rate results can still be extracted.

Assuming that a relationship between $\ln\left(\frac{[\text{VOC}]_0}{[\text{VOC}]}\right)$ and $k$ can be identified, a previously unknown OH rate coefficient, $k_X$, for a specific target VOC, X, can be calculated using multiple known $k$ values as references. An unknown OH rate coefficient will be subject to less uncertainty if its value is interpolated rather than extrapolated and compounds at the extreme

upper and lower limits of the observed relationship will be subject to greater uncertainty. Regardless, the use of multiple known $k$ values should minimise any systematic error induced compared with using a single reference compound.

### 2.1 Choice of reference $k$ values

Some VOCs have literature measurements from two or more laboratories and these can differ considerably. The choice of different reference values for placing the relative rate measurements on an absolute scale can lead to variations in the final

outcome of the experiment. When deciding on reference literature values for use in these experiments, a number of sources were utilised. The IUPAC Task Group on Atmospheric Chemical Kinetic Data Evaluation (http://iupac.pole-ether.fr/) provides recommended values for the reaction between OH and many short chain hydrocarbons, along with some common monoterpene and aromatic VOCs (Atkinson et al., 2006). These values are evaluated using a balance of literature data and are updated regularly and so this data was prioritised. Atkinson and Arey (2003) provide a reviewed dataset for a larger number of VOCs,

collated from several sources, including the extensive evaluations on atmospheric oxidation of alkenes and atmospheric oxidation of aromatic hydrocarbons in Calvert et al. (2000) and Calvert et al. (2002) respectively. In the case that a reference value was not found in the IUPAC Kinetic Data Evaluation, Atkinson and Arey (2003) was used. Reference values at 298 K for only two compounds (1-nonene and 1-octene) could not be found in either the IUPAC evaluated database or Atkinson and Arey (2003). Reference values for these compounds were therefore sourced from Aschmann and Atkinson (2008).

In some cases, the recommendations found in Calvert et al. (2000, 2002) were based upon a single relative rate experiment. The results of those experiments were often based upon outdated recommended rate coefficients for the reference compounds used. As such, and to ensure consistency, the reference values used in this work were updated to reflect any changes in the original relative rate compounds used in those experiments (see Tables 1, 2 and 3 for details).

Evaluated literature reference values for all the compounds used can be found in Tables 1, 2 and 3. VOC depletion

($\ln\left(\frac{[\text{VOC}]_0}{[\text{VOC}]}\right)$) was plotted against these literature values using Eq. (3) and weighted linear regression, using QR decomposition, performed to find the OH exposure ($OH_{exp}$).



## 2.2 Gas sampling and analysis

VOCs emerging from the reactor were collected using a Unity 2 thermal desorption unit (TDU) fitted with a Tenax TA sorbent trap and a CIA 8 Air Server attachment (Markes International). The system was pre-purged at a flow rate of 100 cm$^3$ min$^{-1}$ for six to ten minutes before sampling. During sampling, the trap was maintained below -20 °C and a sampling flow rate of 100 cm$^3$ min$^{-1}$ used for one minute to give a total sample volume of 100 cm$^3$. After sampling, the system lines and sorbent trap were purged with helium carrier gas for three minutes at 100 cm$^3$ min$^{-1}$ to eliminate oxygen from the system before desorption onto the GC column. During the desorption process, the trap was rapidly heated to 250 °C and held for three minutes. All flow paths and sample lines were heated to 150 °C.

The GC system was an Agilent 6890 (Agilent Technologies) fitted with a DB5-MS ultra-inert capillary column (60 m × 0.32 mm ID × 1 μm film, Agilent Technologies) coupled to a Markes International BenchTOF$^{©}$ mass spectrometer. Column head pressure was set to 344 kPa (50 psi) and operated in constant pressure mode using helium as a carrier gas. The temperature ramping of the GC oven was varied between mixtures to achieve optimum separation of VOCs. The GC method and TDU setup gave sample turn arounds of up to 70 minutes for more complex mixtures, and up to 20 minutes for simpler mixtures.

## 2.3 Synthetic Mixtures

Three synthetic gas mixtures were tested using the method outlined above, in which the compounds included in the mixtures, their preparation and the conditions under which they were tested differed. Differences in the reactivity of the VOCs within each mixture were distinguished by calculating the total OH reactivity for that mixture. The OH reactivity for each mixture was calculated as the total sum of each of the reference $k$ values multiplied by the VOC's concentration as shown in Eq. (4).

$$\text{OH reactivity} = \sum k_{\text{VOC}_i+\text{OH}}[\text{VOC}_i] \tag{4}$$

**Mixture 1** was prepared by injecting 1-5 µl of undiluted liquid VOC into a 500 ml evacuated, double ended, stainless-steel sample cylinder (Swagelok). This cylinder was flushed into a pacified gas cylinder (10 L, Experis, Air Products) and filled to approximately 20 bar with N$_2$. The cylinder was then evacuated to atmospheric pressure before being refilled to approximately 100 bar with N$_2$ to achieve a final mixing ratio of each VOC in the cylinder of an estimated 30 parts per billion by volume (ppbv). **Mixture 1** consisted of mainly monoterpenes, along with $m$ and $o$-xylene (Table 1), and was studied at room temperature (average reaction T = 294.5 (± 1.5) K). There was only a small range in OH + VOC rate coefficients and this mixture did not contain any compounds without a literature rate coefficient. The estimated total OH reactivity at standard temperature and pressure (STP) of the VOCs contained in **Mixture 1** was 900 s$^{-1}$. This mixture was diluted with N$_2$ in differing amounts in order to inject gaseous samples into the reactor with a range of OH reactivities between 50 and 300 s$^{-1}$.





**Mixture 2** was prepared using the same method as that used for **Mixture 1** and was also introduced into the reactor at room temperature (294.5 ($\pm$ 1.5) K). The VOCs in this mixture comprised primarily 1-alkenes and cycloalkenes (Table 2) with a range of OH rate coefficients spanning less than a single order of a magnitude. One compound, 2,3-dimethylpent-1-ene, had no reported rate coefficient with OH at room temperature. The estimated total OH reactivity at STP of VOCs contained in

**Mixture 2** was 480 s$^{-1}$. This mixture was diluted with N$_2$ in differing amounts to inject gaseous samples into the reactor with a range of OH reactivities between 25 and 150 s$^{-1}$.

**Mixture 3** was prepared by adding between 1-5 µg, or 1 µl, of each individual solid or liquid VOC to a 25 ml headspace vial. 200 µl of the mixture vapour was then added to a 500 ml evacuated, double ended, stainless steel sample cylinder (Swagelok). This cylinder was flushed into a pacified gas cylinder (Experis, Air Products) and filled to approximately

10 200 bar with ambient air using an oil free, modified RIX compressor (RIX Industries). This mixture contained 43 species, grouped as biogenics, alkenes, alkanes and aromatics (Table 3). The vapour pressure of each compound was used to approximate its concentration in ppbv in the final mixture, to allow for an estimation of the total OH reactivity of the whole mixture of 380 s$^{-1}$ (at STP). To facilitate consistent transmission of low vapour-pressure VOCs through the reactor, **Mixture 3** was introduced at an elevated temperature of 323 ($\pm$ 10) K. This temperature was achieved by wrapping the reactor in heat

tape, and measured using a Type K mineral insulated thermocouple (TCDirect, p/n 408-059) inserted into the reactor.

## 3 Results and Discussion

Synthetic gas mixture results are presented in Sect. 3.1 and 3.2. Results taken from the more complex sampling of ambient air are presented in Sect. 3.3, along with an outline of the adjusted experimental setup. The errors, equal to 1σ (66%), quoted on the measured values in this work are the statistical uncertainties calculated by combining the instrument error and the scatter

in the $\ln\left(\frac{[\text{VOC}]_0}{[\text{VOC}]}\right)$ vs. $k$ data. However, the uncertainties reported in the evaluated literature rate coefficients used as relative rate reference compounds are often large; up to 35% in some cases (Calvert et al., 2002). Using these values for the reference compounds therefore places a limitation on the precision of the results in this work that is not captured by the quoted uncertainties.

### 3.1 Results from relative rate experiments at 295 K

Figure 2 shows sections of typical total ion chromatograms (TIC) obtained for **Mixture 1**; one chromatogram with the reactor lamp turned off (black) and one for a sample with the reactor lamp turned on (blue). When exposed to OH radicals (black), there is a clear reduction in the observed concentration of all VOCs present, reflected in the reduction in peak areas compared with lamp off (blue). The use of selected ion chromatograms provides a clear advantage of using a TOF-MS over a less specific detector due to the reduction in background interference and ease of peak identification. Peak areas were analysed

automatically using peak integration software (Agilent Technologies) with appropriate mass ion selection for each VOC. This allowed for the complete separation of peaks which may otherwise have been unresolved.



To ensure that the observed depletion in each VOCs concentration was entirely due to reaction with OH, experiments were performed in the absence of the OH precursor (i.e. the water bubbler was removed from the system). When the lamp was switched on there was no significant deviation observed in the peak areas for any of the VOC used. As a result, both direct photolysis of VOC and unwanted perturbations due to the lamp (such as heating effects) were deemed to be negligible.

However, other possible perturbations, such as those due to reaction of the VOCs with other photooxidants, could not be ruled out using this method.

Experiments were conducted on each VOC mixture using different mass flow controller settings, allowing for the injection of mixtures into the reactor with five different total VOC loadings, and hence five different total OH reactivities. GC-MS data were analysed using automatic peak integration software and outlying peak areas recommended and removed using

the $MAD_e$ method (Burke, 2001).

Figure 3 shows the relationship between experimentally observed $\ln\left(\frac{[VOC]_0}{[VOC]}\right)$ and literature rate coefficients for **Mixture 1** with an OH reactivity of 180 s$^{-1}$. A clear correlation between $k$ and $\ln\left(\frac{[VOC]_0}{[VOC]}\right)$ can be seen ($R^2 = 0.96$). The nature of Eq. (3) suggests that the fit should be proportional, with an intercept of zero. However, despite the intercept being close to zero, linear fits with a non-zero intercept were found to be a more appropriate approximation. For the calculation to be valid,

there needs to only be a consistent relationship between the depletion of each VOC and $k$. For **Mixture 1** with a low OH reactivity (50 s$^{-1}$), a three-parameter exponential distribution function, given in Eq. (5) was used to fit the data, shown in Fig. 4. Whilst this is inconsistent with Eq. (3), it does not necessarily detract from the relative rate nature of the experiment as a reliable and consistent function can still be plotted through the data. This type of distribution possibly occurs when gas mixtures contain low concentrations of very fast reacting VOCs and arises due to poor mixing conditions within the reactor. As this was

only observed during some experiments, the relative rate data obtained from these plots were disregarded.

$$y = e^{a + \frac{b}{x+c}} \tag{5}$$

The gradient of the line in Fig. 3 is equal to the $OH_{exp}$, the integral of OH concentration over time. $OH_{exp}$ generally decreased

with the increasing OH reactivity of the gas mixture injected (see Supplementary Information Fig. S1). This is in line with other studies by Li et al. (2015) and Peng et al. (2015), who found that increasing the OH reactivity results in an increased rate of removal of OH radicals from the system. The integral of the OH concentration over time is reduced upon increasing the OH reactivity due to the increased number of OH + VOC reactions taking place.

Using Eq. (3) it is possible to estimate new $k$ values at room temperature (295 ($\pm$ 2) K) for all components in **Mixture**

**1** relative to each other. As $OH_{exp}$ is dependent on the OH reactivity of the mixture introduced into the reactor, it is necessary to produce a $k$ value for each VOC at each mass flow controller setting. An error weighted mean of the individual $k$ values determined at each $OH_{exp}$ was then used to assign a final OH rate coefficient for each VOC in the mixture (Table 1).



The majority of VOCs show rates consistent with those in our evaluated literature data set, within experimental error and literature uncertainty. Thus, this technique represents an extension of the classical relative-rate experiment but with the advantage of using multiple species to generate a multivariate, self-consistent relationship between the loss of a VOC and its rate coefficient for reaction with OH, rather than using just one pair of compounds at a time (reactant + reference). By using

multiple reference compounds, the risk of a single erroneous value perturbing the rest of the data is reduced, assuming that all compounds behave in the same way upon exposure to OH.

Whilst the majority of our data is in good agreement with the internally consistent literature data set, the wider literature contains measurements for many of the compounds not included in the IUPAC evaluations or the review paper, Atkinson and Arey (2003). For example, a recent measurement of the oxidation of β-ocimene by OH provided a rate coefficient

of 236 ($\pm$ 54) $\times 10^{-12}$ cm$^3$ molecule$^{-1}$ s$^{-1}$, which is in good agreement with our own result for the same reaction, of 222 ($\pm$ 7) $\times$ $10^{-12}$ cm$^3$ molecule$^{-1}$ s$^{-1}$ (Gaona-Colmán et al., 2016). Recent measurements of the OH + myrcene reaction rate, by Hites and Turner (2009) and Kim et al. (2011) are 50% greater than that calculated in this work. However, their measurements of 335 ($^{+144}_{-101}$) and 334 ($^{+220}_{-132}$) $\times 10^{-12}$ cm$^3$ molecule$^{-1}$ s$^{-1}$ have exceptionally large uncertainties that encompasses the measurement presented in this paper, of 204 ($\pm$ 7) $\times 10^{-12}$ cm$^3$ molecule$^{-1}$ s$^{-1}$. Our measurement of the limonene oxidation rate is also in good

agreement with the recent value in Braure et al. (2014), of 165 ($\pm$ 25) $\times 10^{-12}$ cm$^3$ molecule$^{-1}$ s$^{-1}$.

The measured value for $o$-xylene, of 1.6 ($\pm$ 1.0) $\times 10^{-12}$ cm$^3$ molecule$^{-1}$ s$^{-1}$ is the only result which is not consistent with the literature. It is likely that this is an erroneous measurement that arose due to the OH initiated depletion of $o$-xylene being too small and subject to large relative errors. It was therefore removed from subsequent analyses.

Figure 5 shows an example relative rate plot for **Mixture 2**; for which a linear relationship is observed across all OH

reactivities tested. $k$ values for all the VOCs in this mixture can be estimated using Eq. (3). These values are given in Table 2 and are all consistent with our evaluated literature data set whereas other literature measurements for the OH + cyclohexene and the OH + cyclopentene reactions, of 54 $\times 10^{-12}$ and 45 $\times 10^{-12}$ cm$^3$ molecule$^{-1}$ s$^{-1}$ respectively, are much smaller than our own estimations (Rogers, 1989). Figure 6 demonstrates that our measured rate coefficients for the reactions of OH with 1-alkenes agree well with both experimentally-derived literature rate coefficients and those estimated from structure activity

relationships. Whilst we do not observe a strictly linear trend between $k$ values and increasing carbon chain length, most of our measurements lie between the SAR predictions of Kwok and Atkinson (1995), Peeters et al. (1999, 2007) and Nishino et al. (2009) and are in good agreement with other experimental measurements.

**Mixture 3** contained a much broader range in VOCs, both in terms of functionality and structure, but also in terms of $k$ values. Unlike **Mixtures 1** and **2**, the values for $k$ spanned almost two orders of magnitude; from slow reacting alkanes

such as $n$-heptane, literature $k = 7.2$ ($^{+2}_{-1.6}$) $\times 10^{-12}$ cm$^3$ molecule$^{-1}$ s$^{-1}$, to faster reacting monoterpenes such as limonene, literature $k = 170$ ($\pm$ 51) $\times 10^{-12}$ cm$^3$ molecule$^{-1}$ s$^{-1}$ (Atkinson and Arey, 2003). The concentrations of the individual VOCs in the mixture were also much lower compared with the other mixtures. Both the complexity of the mixture and the low concentration of the constituents make this mixture more representative of ambient atmospheric conditions. This mixture was studied at an elevated



reactor temperature of $T = 323 \ (\pm 10)$ K in order to prevent excessive partitioning of the less-volatile VOCs to the reactor walls.

Figure 7 shows the results of a relative rate experiment for **Mixture 3**. Only compounds with a known rate coefficient at 323 K are shown. The relationship between VOC depletion ($\ln\left(\frac{[VOC]_0}{[VOC]}\right)$) and $k$ shows an exponential cumulative distribution,

likely due to the small concentrations of VOCs injected and the large range in VOC + OH rate coefficients. The inset shows that it is possible to approximate the rate of VOC + OH for species with a $k$ value (at 323 ($\pm 10$) K) of less than $30 \times 10^{-12}$ cm$^3$ molecule$^{-1}$ s$^{-1}$ using linear regression. Using this method, the estimated $k$ values for many of the VOCs in **Mixture 3** are consistent with those in the literature (Table 3), suggesting that measurements at close to ambient atmospheric conditions are possible. Two measurements stand out as contrasting with their literature values; that for myrcene, of $310 \ (\pm 105) \times 10^{-12}$ cm$^3$

molecule$^{-1}$ s$^{-1}$ and that for β-ocimene, of $950 \ (\pm 800) \times 10^{-12}$ cm$^3$ molecule$^{-1}$ s$^{-1}$. These two compounds were at the higher end of the reactivity scale in **Mixture 3**, and hence small deviations in their depletion would lead to large variations in their measured values using the derived curved relationships. This is reflected in the comparatively large errors relative to the other VOCs in the mixture. Similar to *o*-xylene in **Mixture 1**, the results for these two compounds were removed from subsequent analyses.

Figure 8 shows an overall plot of all measured rate coefficients taken from **Mixtures 1, 2** and **3** plotted against their literature reference counterparts. Solid data points are for measurements made at 294 ($\pm 2$) K whilst empty data points are for measurements conducted at the elevated reactor temperature of 323 ($\pm 10$) K. The black solid line represents a 1:1 fit. Linear regression may be performed on the dataset in its entirety to determine the overall ability of this method for replicating literature rate coefficients. The gradient of this regression is 1.0 which suggests that this experimental technique is able to replicate the

literature exceptionally well. The vast majority of data lie well within an uncertainty in the literature of 25%, shown by the grey shaded area. Outliers include: 1-hexene at room temperature and 2,2,3-trimethylbutane, n-nonane and β-pinene at 323 K.

Our studies suggest that a linear response between $\ln\left(\frac{[VOC]_0}{[VOC]}\right)$ and $k$ is best achieved when the VOCs in a mixture have a small range in OH rate coefficients (i.e. they react with OH at similar rates, with respect to each other). It appears that a non-linear response occurs when the mixing of VOCs and radicals are on similar timescales to the chemistry. Therefore,

mixtures that satisfy the above criteria are preferred for the estimation of novel rate coefficients, as demonstrated using **Mixture 2**. Whilst there is still a good correlation between $\ln\left(\frac{[VOC]_0}{[VOC]}\right)$ and $k$ for mixtures which do not satisfy this criterion, the non-linearity makes the estimation of novel rate coefficients more challenging but still possible. These mixtures may therefore be better suited for highlighting inconsistencies within the current literature data set for certain groups of compounds.

### 3.2 Determination of new rate coefficients

**Mixture 2** contained the compound 2,3-dimethylpent-1-ene, for which we could find no literature rate coefficient measurements at the time of writing. 2,3-dimethylpent-1-ene has been detected in emissions from certain tomato variants



cultivated in Portugal (Figueira et al., 2014). Using Eq. (3), it is possible to estimate an experimentally derived room temperature OH rate coefficient for this compound of $k = 57$ ($\pm 3$) $\times 10^{-12}$ cm$^3$ molecule$^{-1}$ s$^{-1}$. As discussed earlier in Sect. 3, the error in this value is the statistical uncertainty calculated by combining the instrument error and the scatter in the data. However, the calculation of measurement uncertainty does not reflect the larger uncertainties, of up to 35%, in the rate

coefficients used as reference compounds and can therefore be considered an underestimation (lower limit) of the true uncertainty.

SARs are often used to predict rate coefficients for those compounds which do not have experimentally determined values. Three distinct SARs in the literature can be used to estimate a rate coefficient for 2,3-dimethylpent-1-ene. Using the methods outlined in Kwok and Atkinson (1995), Peeters et al. (1999, 2007) and Nishino et al. (2009), the rate coefficient for

2,3-dimethylpent-1-ene can be estimated as $k = 55.0$, 63.1 and 59.3 $\times 10^{-12}$ cm$^3$ molecule$^{-1}$ s$^{-1}$ respectively. These are all in good agreement with the experimental measurement made in this work.

Using the same method for **Mixture 3,** $k$ values at 323 K for 20 compounds, as yet unmeasured at this temperature, can be calculated (Table 3). However, the straight line becomes increasingly less appropriate for estimating new rate coefficients at higher VOC depletion as the linear regression begins to deviate further from the data. Table 3 therefore also

provides rate coefficients for all the compounds in **Mixture 3** calculated using the equation of a curve, as opposed to the equation of the straight line, as shown in the inset of Fig. **7**. This demonstrates that it is possible to derive new rate coefficients providing that there is a consistent relationship between VOC depletion and rate coefficient. In this way, using the curve, rate coefficients for a further 5 compounds with large depletions ($k > 30 \times 10^{-12}$) are estimated.

For the OH + monoterpene reactions, the rate coefficients determined in this work for **Mixture 3** are in good

agreement with rate coefficients derived by others at this temperature, within experimental errors and literature uncertainty. This demonstrates that this type of experiment is capable of assessing the temperature dependence of VOC + OH rate coefficients. The 323 K rate coefficients for the OH + alkene reactions consistently show a non-Arrhenius temperature dependence when compared with their room temperature counterparts. With the exception of 1-heptene, most of the estimated coefficients for OH + alkene reactions at 323 K are within 20% of the 298 K literature values; 1-heptene displays an

anomalously low result. This reflects the non-Arrhenius relationship described and observed in the literature for OH addition to alkenes and monoterpenes (Chuong et al., 2002; Kim et al., 2011).

The rate coefficients for OH + branched alkanes show a positive temperature dependence, which contrasts with the straight and cyclic alkanes. We are also able to derive 323 K rate coefficients for the reaction between OH and three alkanes (2-methylheptane, 3-methylheptane and ethylcyclohexane) for which we could find no 298 K data in the literature.

Rate coefficients for the reaction between OH and 12 aromatic VOC at 323 K are also estimated for the first time. There appears to be little to no correlation with temperature; some estimates are lower and some higher than the 298 K literature values. 3-ethyltoluene is a possible anomaly in the series, with a 323 K rate coefficient almost twice that of the 298 K literature measurement. This, however, highlights the need for relevant temperature dependent rate coefficient data.



### 3.3 Determination of rate coefficients by ambient air sampling

The use of synthetic gas mixtures places a limitation on the complexity of mixtures that can be introduced into the reactor and can only include compounds for which authentic standards or pure raw materials are available. The atmosphere represents an upper limit of mixture complexity, with regards to the VOC matrix, with the added benefit of containing VOCs which may
not be readily available or readily synthesised, such as secondary or tertiary oxidation products. Should the determination of rate coefficients through ambient atmospheric sampling be possible, comprehensive measurements of atmospherically relevant VOC + OH reactions could be made.

Unfortunately, there are practical challenges associated with real atmospheric air sampling. First, the complexity of the atmosphere itself poses a problem and, whilst a mass spectrometer may aid with the identification of VOCs, single column
gas chromatography may not always be enough to fully resolve all of the many individual compounds observed. The presence of additional oxidants and nitrogen oxides ($NO_x$) may also lead to further complications and interfere with the VOCs concentrations during sampling and measurement. Many VOCs also exist in the atmosphere in concentrations that are orders of magnitude lower than those used in the synthetic gas mixtures outlined above, meaning that detection may be a problem, particularly after depletion using synthetic oxidants. There is also likely to be a large variety in the concentrations at which
atmospheric VOCs exist, which could have an adverse effect on the uniformity of individual VOC exposure to OH. Finally, the atmosphere is a dynamic mixture; the components and their concentrations are subject to constant change and as such, the experimental technique would need to account for these, often rapid, temporal fluctuations.

Some of these issues can be overcome by using a dual reactor instrumental set up, allowing for two ambient air samples to be sampled simultaneously, as shown in Fig. 9. If OH is absent in both reactors then identical samples should be
observed. However, in practice, it is impossible to obtain two completely indistinguishable VOC measurements due to random but minor variations in sampling and later analysis. However, comparisons between the two reactors over time can be established to account for any systematic differences that are observed between the two systems.

OH initiated reactions can be induced in one of the reactors whilst the other is sampled simultaneously in the absence of synthetic oxidants. This allows for the construction of a dataset of VOC concentrations with and without synthetic OH
radicals present. Unlike for synthetic mixtures, where VOC concentrations were averaged before processing using Eq. (3), the natural variability in atmospheric VOC mixing ratios dictates that a $\ln\left(\frac{[\text{VOC}]_0}{[\text{VOC}]}\right)$ value must be calculated for each VOC for each dual sample.

Using this experimental technique for ambient air analysis, VOC reactivity was analysed at the Wolfson Atmospheric Chemistry Laboratories at the University of York, a suburban site in York, United Kingdom during the period 6th -10th June
2014. The sampling site was approximately 200 m from a local commuter road, 1.5 km from a dual carriageway and in close proximity to a small area of woodland consisting primarily of oak and sycamore trees. There was a large dynamic range in the concentrations of VOC observed; some species, for instance styrene, experienced large degrees of scatter, due in part to very low ambient mixing ratios (16 parts per trillion by volume (pptv)). 5 ppb of ozone was also detected at the end of the reactor.




This could be attributed to the presence of non-synthetic oxidants and $NO_x$, but is likely to be the ambient ozone sampled from outside that made it through the heated sample lines. At this concentration, given the rate of reaction between $O_3$ and all the compounds analysed, it is unlikely that the $O_3$ would have a significant impact on the degradation of any species within the reactor. Indeed, even for the reaction between $O_3$ and styrene, which proceeds at $1.70 \times 10^{-17}$ cm$^3$ molecule$^{-1}$ s$^{-1}$ (Atkinson and

5  Arey, 2003), the depletion by $O_3$ accounts for less than one per cent of its total depletion (see Supplementary Information Table S2).

23 VOCs (including alkanes, alkenes, aromatics, isoprene, monoterpenes, naphthalene, ethyl acetate, dichloromethane – see Table 4) were positively identified from the GC-MS data. The vast majority of identified VOCs have very slow rate coefficients for reaction with $O_3$, with a maximum of $1 \times 10^{-20}$ cm$^3$ molecule$^{-1}$ s$^{-1}$ for the alkanes and

alkylbenzenes (Atkinson and Arey, 2003). Of the 23 identified compounds, four monoterpenes (α-pinene, β-pinene, 3-carene and limonene) react relatively fast with both OH and $O_3$ (see Table 4) and their concentrations were decreased to below the limit of detection for the "lamp on" samples. These four compounds were therefore removed from subsequent analyses.

Depletions for the eight identified compounds which have 323 K literature rate coefficients are shown in Fig. 10 in blue. There is a clear linear relationship, shown by the black dashed line, between the depletion in these eight compounds and

their rate of reaction with OH. However, a significant section of the plot lacks literature data for comparison. To aid in the clarification of the linear relationship between rate coefficient and depletion, seven compounds, whose rate coefficients were derived in earlier studies using synthetic gas mixtures (**Mixture 3**) are also plotted in yellow. Six of these compounds are in good agreement with the linear correlation derived using the literature, within experimental error. A linear fit to the 323 K data which excludes isoprene, the fasting reacting observed VOC, is shown by the red dashed line. This linear regression gives a

slightly better fit to the data derived from the synthetic gas mixture.

The relative rate results for 19 compounds are provided in Table 4. Except for n-hexane and dichloromethane, the 323 K rate coefficients for six compounds (benzene, isoprene, *n*-heptane, *n*-decane, naphthalene and toluene) are in reasonable agreement to comparable rate coefficients in the literature. Two compounds (dichloromethane and ethyl acetate) yield negative rate coefficients as the depletions in their concentrations were generally smaller than the instrument noise. New rate

coefficients, at 323 K, for three compounds (dodecane, styrene and tridecane) are also derived for the first time.

The ambient air derived rate coefficient for the OH + 3-ethyltoluene reaction is almost twice as large as that determined using the synthetic mixture, which itself was anomalously larger than the 298 K literature equivalent. Many *k* values determined using ambient air sampling are systematically larger than those in the literature for reactions at 298 K. This is likely due to the increased complexity of sampling and analysing ambient air when compared with synthetic gas mixtures

(as discussed above) and the large range in terms of both reactivity and concentration in the compounds sampled. However, it is clearly still possible to estimate reasonable rate coefficients, albeit with higher uncertainties, for most of the ambient observed compounds. The use of this technique is therefore especially useful for VOCs which are not commercially available or easily synthesised or isolated, such as the oxidation products of common atmospheric species.



## 4 Atmospheric implications and conclusions

We have demonstrated here a simple and versatile method for measuring relative rate coefficients for reactions of OH with multiple VOCs simultaneously, using the well-established relative rate kinetic technique as a basis. The method builds upon that used to estimate the contribution of unidentified VOCs to OH reactivity (Kato et al., 2011). Three synthetic gas mixtures have been used with this technique, covering a broad range in VOC functionality and in rates of reaction with OH radicals. Additionally, these mixtures have been studied at different temperatures, showing that this method could be applied to quickly generate temperature dependant Arrhenius expressions for multiple VOCs with only very simple, minor modifications to the current set up. Consequently, we have been able to estimate novel rate coefficients at 323 K for a total of 28 atmospherically relevant VOCs, including three that have no measurements at 298 K (2-methylheptane, 3-methylheptane and ethylcyclohexane). We have also been able to provide results which are in good agreement with the only previously recorded measurements for the reactions between OH and γ-terpinene and OH and cycloheptene, along with various other VOCs for which few results were previously available.

We have shown that this method can produce novel results by providing the first measurement of the OH + 2,3-dimethylpent-1-ene reaction, at room temperature, of $k = 5.7\ (\pm\ 0.3) \times 10^{-11}$ cm$^3$ molecule$^{-1}$ s$^{-1}$. Using an approximate atmospheric concentration of OH, of $1 \times 10^6$ molecules cm$^{-3}$, this gives an estimated OH lifetime for 2,3-dimethylpent-1-ene of 4.9 hours. This corresponds to 2.5 times its loss rate with respect to O$_3$ (estimated using an approximate atmospheric O$_3$ concentration of 30 ppb and the SAR provided in King et al. (1999)).

We have also shown that using a dual-reactor set up allows for the derivation of rate coefficients via ambient air sampling. Rate coefficients for six compounds were shown to be in reasonable agreement with the literature. Additionally, a further three new rate coefficients were measured at 323 K using this method of ambient air sampling (dodecane, styrene and tridecane). Further improvements in analytical detection limits and chromatographic resolution would allow the method to be extended to more VOC, including less volatile organics. By using a TOF-MS, there is also the possibility of retrospectively determining OH + VOC reaction rates, should a species be deemed important in the future.

In summary, this technique represents a significant breakthrough in the measurement of atmospherically relevant oxidation kinetics. The throughput of target compounds has been improved dramatically with respect to the base relative rate technique, which will allow for the rapid development of experimentally derived measurements for homologous series of multifunctional, long chain and branched systems, as demonstrated by that for the 1-alkenes in this work. The possibility of conducting temperature dependent studies will allow for a significant broadening of our knowledge in this area. The technique could also be adapted for use with Cl or NO$_3$ radicals, for which oxidative kinetic measurements are severely limited in the literature. Additionally, the use of ambient air with this technique demonstrates its versatility, and highlights the possibility of measuring reaction rates for difficult to synthesise, or currently unidentified, VOCs. Finally, the rate coefficients for reactions of OH with unidentified VOCs in the atmosphere can be measured using this method. If the concentrations of these species can be estimated, then the method would enable missing OH reactivity to be quantified.



**Data availability**

Raw data is available upon request.

**Author contribution**

ACL, ARR, DEH, JFH & LKW planned the overall project; DEH, LJC, JFH & TJD designed experiments; DRC, GAB, LKW,
NR, RED & TI conducted initial experiments; JTS & RTL conducted later experiments; JTS & RTL performed the analysis;
JTS, RTL & TJD prepared the manuscript with contributions from all authors.

**Competing interests**

The authors declare that they have no conflict of interest.

**Acknowledgements**

The authors would like to thank Mat Evans for his contribution to an earlier iteration of this work, Martyn Ward for his
excellent technical support throughout the project, Abigail Mortimer, Chris Mortimer and Chris Rhodes for their assistance in
the respective glass, mechanical and electronic workshops. We are grateful to the Natural Science Research Council (NERC)
for funding this work under grant numbers NE/I012737/1, NE/I014616/1 and NE/J008532/1. We would also like to thank
NERC for the provision of a research studentship (DRC) and the funding of a PhD as part of the SPHERES DTP scheme
(JTS).

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



**Table 1 Results of relative rate experiments, with literature data, for each VOC in Mixture 1, ordered by measured $k$ value.**

| Compound | Range of depletion / % | Measured $k$ (295 K)[a] / $10^{-12}$ cm$^3$ molecule$^{-1}$ s$^{-1}$ | Evaluated literature $k$ (298 K) / $10^{-12}$ cm$^3$ molecule$^{-1}$ s$^{-1}$ | Reference[b] | Number of literature measurements |
|---|---|---|---|---|---|
| β-ocimene | 27-70 | 222 ± 7 | 245 ± 49[c] | Atkinson and Arey, 2003 | 4 |
| γ-terpinene | 24-63 | 206 ± 6 | 170 ($^{+44}_{-35}$) | Atkinson et al., 2006[d] | 1 |
| myrcene | 24-62 | 204 ± 7 | 209 ± 42[c] | Atkinson and Arey, 2003 | 4 |
| limonene | 18-57 | 152 ± 4 | 170 ± 51 | Atkinson and Arey, 2003 | 5 |
| isoprene | 12-41 | 104 ± 6 | 100 ($^{+15}_{-13}$) | Atkinson et al., 2006[d] | 25+ |
| 3-carene | 11-40 | 97 ± 3 | 85 ± 17[c] | Atkinson and Arey, 2003 | 2 |
| β-pinene | 9-34 | 76 ± 9 | 79 ± 20 | Atkinson and Arey, 2003 | 10 |
| α-pinene | 7-24 | 56 ± 6 | 53 ($^{+22}_{-15}$) | Atkinson et al., 2006[d] | 9 |
| m-xylene | 3-13 | 21 ± 6 | 23 ± 4 | Atkinson and Arey, 2003 | 15 |
| o-xylene | 2-7 | 1.6 ± 1.0 | 13 ± 3 | Atkinson and Arey, 2003 | 10 |

[a] errors are likely underestimated as they do not include the large uncertainties in the literature rate coefficients used as reference compounds.

[b] data from Atkinson and Arey (2003) is collated from multiple sources including Calvert et al. (2000, 2002).

[c] original result relative to 2,3-dimethylbut-2-ene; $k = 1.13 \times 10^{-10}$ cm$^3$ molecule$^{-1}$ s$^{-1}$. Updated result relative to $k = 1.1$ (± 0.2) $\times 10^{-10}$ cm$^3$ molecule$^{-1}$ s$^{-1}$ as recommended in Calvert et al. (2002).

[d] see also IUPAC Task Group on Atmospheric Chemical Kinetic Data Evaluation Website – http://iupac.pole-ether.fr



**Table 2 Results of relative rate experiments, with literature data, for each VOC in Mixture 2, ordered by measured $k$ value.**

| Compound | Range of depletion / % | Measured $k$ (295 K)[a] / $10^{-12}$ cm$^3$ molecule$^{-1}$ s$^{-1}$ | Evaluated literature $k$ (298 K) / $10^{-12}$ cm$^3$ molecule$^{-1}$ s$^{-1}$ | Reference[b] | Number of literature measurements |
|---|---|---|---|---|---|
| isoprene | 8-26 | 101 ± 4 | 100 ($\pm^{5}_{3}$) | Atkinson et al., 2006[d] | 25+ |
| β-pinene | 5-21 | 74 ± 12 | 79 ± 20 | Atkinson and Arey, 2003 | 10 |
| cycloheptene | 6-20 | 74 ± 9 | 74 ± 10[c] | Atkinson and Arey, 2003 | 1 |
| cyclohexene | 6-20 | 71 ± 4 | 68 ± 17 | Atkinson and Arey, 2003 | 9 |
| cyclopentene | 7-19 | 67 ± 8 | 67 ± 23 | Atkinson and Arey, 2003 | 3 |
| 2,3-dimethylpent-1-ene | 5-17 | 57 ± 3 | | | |
| α-pinene | 4-17 | 53 ± 4 | 53 ($\pm^{22}_{15}$) | Atkinson et al., 2006[d] | 9 |
| 1-octene | 4-15 | 43 ± 5 | 41.4 ± 0.8 | Aschmann and Atkinson, 2008 | 2 |
| 1-nonene | 4-15 | 42 ± 3 | 43.2 ± 0.5 | Aschmann and Atkinson, 2008 | 2 |
| 1-heptene | 3-13 | 35 ± 3 | 40 ± 12 | Atkinson and Arey, 2003 | 2 |
| 1-hexene | 2-15 | 29 ± 21 | 37 ± 11 | Atkinson and Arey, 2003 | 2 |

[a] errors are likely underestimated as they do not include the large uncertainties in the literature rate coefficients used as reference compounds.

[b] data from Atkinson and Arey (2003) is collated from multiple sources including Calvert et al. (2000, 2002).

5    [c] original result relative to isoprene; $k = 1.01 \times 10^{-10}$ cm$^3$ molecule$^{-1}$ s$^{-1}$. Updated result relative to $k = 1.00$ ($\pm^{15}_{13}$) $\times 10^{-10}$ cm$^3$ molecule$^{-1}$ s$^{-1}$ as recommended by Atkinson et al. (2006)[§].

[d] see also IUPAC Task Group on Atmospheric Chemical Kinetic Data Evaluation website – http://iupac.pole-ether.fr




**Table 3 Results of relative rate experiments, with literature data, for each VOC in Mixture 3, ordered by curve measured $k$ value.**

| Group | Compound | $p_0{}^a$ / mmHg | Estimated cylinder concentration / ppb | Linear measured $k$ (323 K) / $10^{-12}$ cm$^3$ molecule$^{-1}$ s$^{-1}$ | Curve measured $k$ (323 K) / $10^{-12}$ cm$^3$ molecule$^{-1}$ s$^{-1}$ | Literature $k$ / $10^{-12}$ cm$^3$ molecule$^{-1}$ s$^{-1}$ 323 K | Evaluated literature $k$ / $10^{-12}$ cm$^3$ molecule$^{-1}$ s$^{-1}$ 298 K |
|---|---|---|---|---|---|---|---|
| **Biogenics** | β-ocimene | 1.60 | 0.23 | | 950 ± 800 | 261 ($^{+54}_{-48}$)[e] | 245 ± 49[b] |
| | myrcene | 2.09 | 0.31 | | 310 ± 105 | 253 ($^{+68}_{-56}$)[d] | 209 ± 42[b] |
| | γ-terpinene | 1.10 | 0.16 | | 180 ± 19 | | 170 ($^{+44}_{-35}$)[c] |
| | limonene | 1.98 | 0.29 | | 140 ± 5 | 148 ($^{+21}_{-18}$)[b] | 170 ± 51[b] |
| | 3-carene | 1.90 | 0.28 | | 81 ± 1.5 | | 85 ± 17[b] |
| | β-pinene | 2.93 | 0.43 | | 81 ± 20 | 66 ($^{+11}_{-9.4}$)[1] | 79 ± 20[1] |
| | isoprene | 550 | 80.4 | | 75 ± 8 | 90 ($^{+33}_{-24}$)[2] | 100 ($^{+15}_{-13}$)[2] |
| | α-pinene | 4.75 | 0.69 | 48 ± 3 | 50 ± 2 | 47 ($^{+8.3}_{-7.0}$)[2] | 53 ($^{+22}_{-15}$)[2] |
| | camphene | 2.50 | 0.37 | 47 ± 3 | 50 ± 2 | | 53 ± 7[1] |
| **Alkenes** | cycloheptene | 22.5 | 3.29 | | 71 ± 4 | | 74 ± 10[b] |
| | cyclopentene | 378 | 55.3 | 52 ± 2 | 58 ± 0.9 | | 67 ± 23[1] |
| | cyclohexene | 89.0 | 13.0 | 52 ± 5 | 57 ± 4 | | 68 ± 17[b] |
| | 1-hexene | 184 | 26.9 | 32 ± 6 | 31 ± 5 | | 37 ± 11[1] |
| | 1-heptene | 59.3 | 8.67 | 17 ± 2 | 18 ± 1.2 | | 40 ± 12[b] |
| **Alkanes** | cyclooctane | 4.60 | 0.67 | 17 ± 0.8 | 17 ± 0.5 | 15 ($^{+8}_{-5}$)[b] | 13 ± 0.4[b] |
| | cycloheptane | 21.6 | 3.16 | 14 ± 2 | 15 ± 1.2 | 13 ($^{+6}_{-4}$)[b] | 12 ± 3[b] |
| | ethylcyclohexane | 31.0 | 4.54 | 15 ± 1.0 | 15 ± 0.5 | | |
| | 3-methylheptane | 19.6 | 2.87 | 12 ± 3 | 14 ± 2 | | |
| | *n*-decane | 1.43 | 0.21 | 13 ± 3 | 14 ± 1.5 | 12 ($^{+2}_{-1.9}$)[b] | 11 ± 2[b] |
| | *n*-nonane | 4.45 | 0.65 | 14 ± 1.4 | 14 ± 0.7 | 10 ($^{+1.6}_{-1.0}$)[b] | 10 ± 1.9[b] |
| | 2-methylheptane | 20.5 | 3.00 | 12 ± 0.2 | 12 ± 0.2 | | |





| | | | | | | |
|---|---|---|---|---|---|---|
| 2-methylpentane | 211 | 30.8 | 10 ± 3 | 12 ± 2 | | 5.2 ± 1.3[1] |
| 3-methylpentane | 190 | 27.8 | 8.4 ± 1.2 | 9.1 ± 0.9 | | 5.2 ± 1.3[1] |
| *n*-octane | 14.1 | 2.06 | 6.8 ± 2 | 8.8 ± 1.4 | 8.7 ($^{+2}_{-1.7}$)[1] | 8.1 ± 1.6[1] |
| *n*-heptane | 46.0 | 6.73 | 8.1 ± 0.8 | 7.9 ± 0.5 | 7.2 ($^{+2}_{-1.6}$)[1] | 6.8 ± 1.4[1] |
| 2,2,3-trimethylbutane | 90.0 | 13.2 | 3.8 ± 1.2 | 6.3 ± 2 | 4.0 ($^{+1.7}_{-1.2}$)[1] | 3.8 ± 1.0[1] |
| **Aromatics** 3-ethyltoluene | 3.00 | 0.44 | 34 ± 0.9 | 34 ± 0.6 | | 19 ± 7[b] |
| 1,2,4-trimethylbenzene | 2.10 | 0.31 | 32 ± 1.5 | 32 ± 1.0 | | 33 ± 8[b] |
| 1,2,3-trimethylbenzene | 1.69 | 0.25 | 31 ± 1.2 | 31 ± 0.9 | | 33 ± 8[b] |
| naphthalene | 0.085 | 0.01 | 22 ± 1.4 | 22 ± 1.2 | 22.4[b] | 23 ± 6[b] |
| m-xylene | 8.29 | 1.21 | 21 ± 1.5 | 21 ± 1.3 | | 23 ± 4[1] |
| indane | 1.50 | 0.22 | 19 ± 0.6 | 19 ± 0.6 | | 19 ± 8[b] |
| 4-ethyltoluene | 2.90 | 0.42 | 16 ± 0.4 | 16 ± 0.14 | | 12 ± 4[b] |
| 2-ethyltoluene | 2.60 | 0.38 | 15 ± 1.0 | 15 ± 0.5 | | 12 ± 4[b] |
| o-xylene | 6.61 | 0.97 | 12 ± 1.0 | 13 ± 0.6 | | 13 ± 3[1] |
| 4-isopropyltoluene | 1.50 | 0.22 | 12 ± 1.2 | 12 ± 0.9 | | 14 ± 3[b,f] |
| *n*-propylbenzene | 4.50 | 0.66 | 8.5 ± 0.7 | 8.5 ± 0.5 | | 5.8 ± 1.5[1] |
| isopropylbenzene | 4.50 | 0.66 | 5.6 ± 1.5 | 7.2 ± 1.0 | | 6.3 ± 2[1] |
| ethylbenzene | 9.60 | 1.40 | 5.7 ± 1.3 | 7.0 ± 1.0 | | 7.0 ± 2[1] |
| toluene | 28.4 | 4.15 | 4.9 ± 0.6 | 5.1 ± 0.3 | 5.2 ($^{+4}_{-2}$)[2] | 5.6 ($^{+1.5}_{-1.3}$)[2] |

[a] *p₀* refers to the vapour pressure of the VOC. This was used to estimate the concentration of VOC transferred to the cylinder.

[b] Atkinson and Arey, 2003

[c] Atkinson et al., 2006 (see also IUPAC Task Group on Atmospheric Chemical Kinetic Data Evaluation website – http://iupac.pole-ether.fr)

[d] Hites and Turner, 2009

[e] Kim et al., 2011

[f] original result relative to cyclohexane; $k = 7.14 \times 10^{-12}$ cm³ molecule⁻¹ s⁻¹. Updated result relative to $k = 6.97$ (± 1.4) $\times 10^{-12}$ cm³ molecule⁻¹ s⁻¹ as recommended by Atkinson and Arey (2003).



**Table 4 Results of relative rate experiments, with literature data, for VOC in the ambient air analysis, ordered by ambient air measured $k$ value.**

| compound | ambient air measured $k$ (323 K) / $10^{-12}$ cm$^3$ molecule$^{-1}$ s$^{-1}$ | synthetic mixture measured $k$ (323 K) / $10^{-12}$ cm$^3$ molecule$^{-1}$ s$^{-1}$ | literature $k$ (323 K) / $10^{-12}$ cm$^3$ molecule$^{-1}$ s$^{-1}$ | literature $k$ (298 K) / $10^{-12}$ cm$^3$ molecule$^{-1}$ s$^{-1}$ |
|---|---|---|---|---|
| tridecane | 140 ± 10 | | | 15.1 ± 3.8[a] |
| isoprene | 90 ± 3 | 75 ± 8 | 90 ($^{+33}_{-24}$)[b] | 100 ($^{+15}_{-13}$)[b] |
| styrene | 81 ± 7 | | | 58 ± 12[a] |
| 3-ethyltoluene | 70 ± 5 | 34 ± 0.6 | | 19 ± 7[a] |
| dodecane | 58 ± 8 | | | 13 ± 3[a] |
| 1,2,3-trimethylbenzene | 44 ± 4 | 31 ± 0.9 | | 33 ± 8[a] |
| 1,2,4-trimethylbenzene | 40 ± 4 | 32 ± 1.0 | | 33 ± 8[a] |
| naphthalene | 29 ± 5 | 22 ± 1.2 | 22.4[a] | 23 ± 6[a] |
| 2-ethyltoluene | 24 ± 3 | 15 ± 0.5 | | 12 ± 4[a] |
| *n*-decane | 20 ± 3 | 14 ± 1.5 | 12 ($^{+2}_{-1.9}$)[a] | 11 ± 2[a] |
| ethylbenzene | 16 ± 2 | 7.0 ± 1.0 | | 7.0 ± 2[a] |
| *n*-heptane | 9.3 ± 2 | 7.9 ± 0.5 | 7.2 ($^{+2}_{-1.6}$)[a] | 6.8 ± 1.4[a] |
| *n*-propylbenzene | 7.4 ± 3 | 8.5 ± 0.5 | | 5.8 ± 1.5[a] |
| toluene | 5.4 ± 1.2 | 5.1 ± 0.3 | 5.2 ($^{+4}_{-2}$)[a] | 5.6 ($^{+1.5}_{-1.2}$)[b] |
| benzene | 3.6 ± 1.1 | | 1.3 ± 0.4[a] | 1.2 ± 0.2[a] |
| isopropylbenzene | 1.8 ± 0.8 | 7.2 ± 1.0 | | 5.8 ± 1.2[a] |
| *n*-hexane | 0.86 ± 0.5 | | 5.8 ± 0.5[a] | 5.2 ± 1.3[a] |
| ethyl acetate | -0.13 ± 0.04 | | | 1.7 ± 0.3[c] |
| dichloromethane | -1.5 ± 0.9 | | 0.13 ($^{+0.07}_{-0.05}$)[b] | 0.1 ($^{+0.03}_{-0.02}$)[b] |

[a] Atkinson and Arey, 2003

[b] Atkinson et al., 2006 (see also IUPAC Task Group on Atmospheric Chemical Kinetic Data Evaluation website –

5   http://iupac.pole-ether.fr)

[c] Calvert et al., 2011



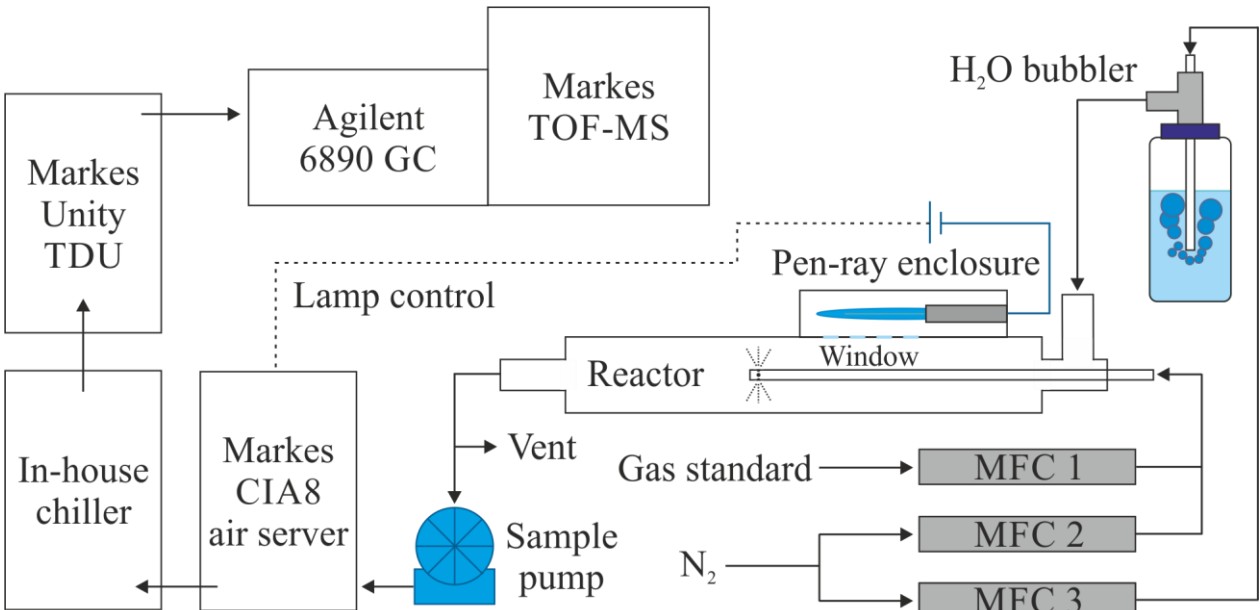

**Figure 1 Schematic of the OH reactor configuration used. Key to abbreviations: CIA8 = air server and canister interface accessory; GC = gas chromatograph; MFC = mass flow controller; TOF-MS = time of flight mass spectrometer; TDU = thermal desorption unit. The flow rate through MFC 1 was stepped from 200 sccm through to 1000 sccm in 200 sccm intervals. The combined flow rate through MFC 1 and MFC 2 was kept constant at 1000 sccm. The flow rate through MFC 3, and hence through the H$_2$O bubbler, was set to 2000 sccm resulting in a total flow through the reactor of 3000 sccm. The residence time of VOC inside the reactor after injection was approximately 4 s, with the oxidation chemistry expected to occur in under 0.5 s.**



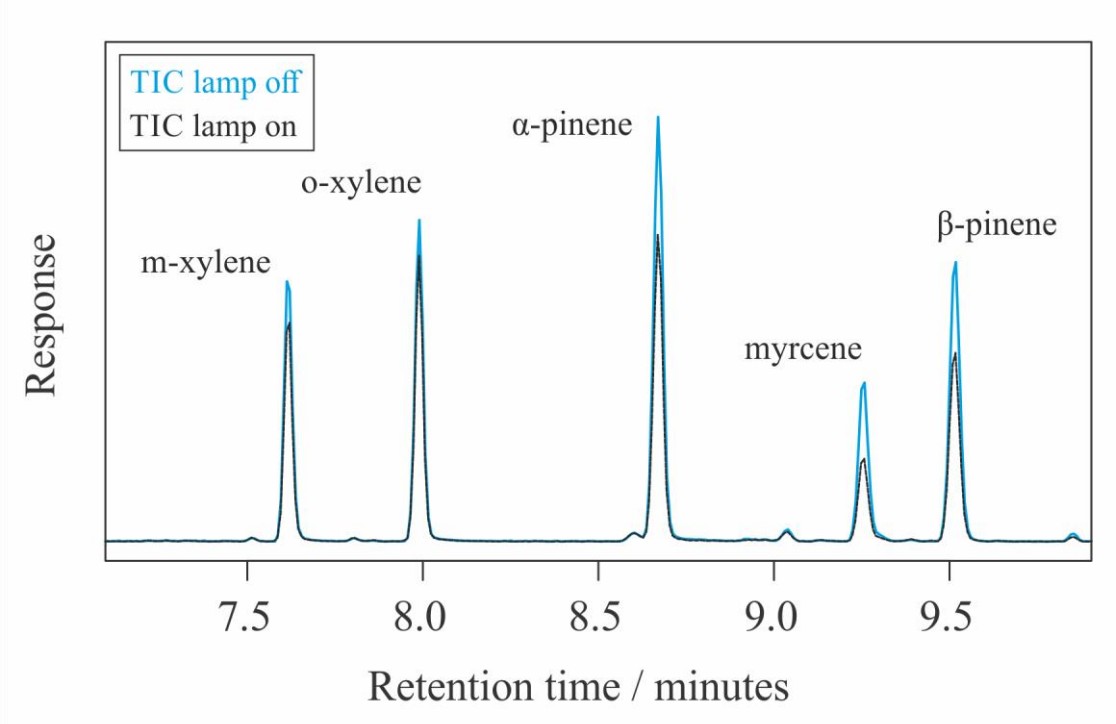

**Figure 2 Typical total ion chromatograms (TICs) obtained for a ppbv mixing ratio gas standard (Mixture 1) with the lamp off (black) and lamp on (blue). Greater differences in peak areas are observed for VOC which react faster with OH. Literature rate coefficients (in units of $10^{-12}$ cm$^3$ molecule$^{-1}$ s$^{-1}$) for the VOC are m-xylene[1], 23 (± 4); o-xylene[1], (13 ± 3); α-pinene[2], (53 $^{+22}_{-15}$); myrcene[1], 209 ± (42) and β-pinene[1], 79 ± (20).**

[1] Atkinson and Arey (2003) updated to reflect changes in reference compound
[2] Atkinson et al. (2006) – http://iupac.pole-ether.fr/



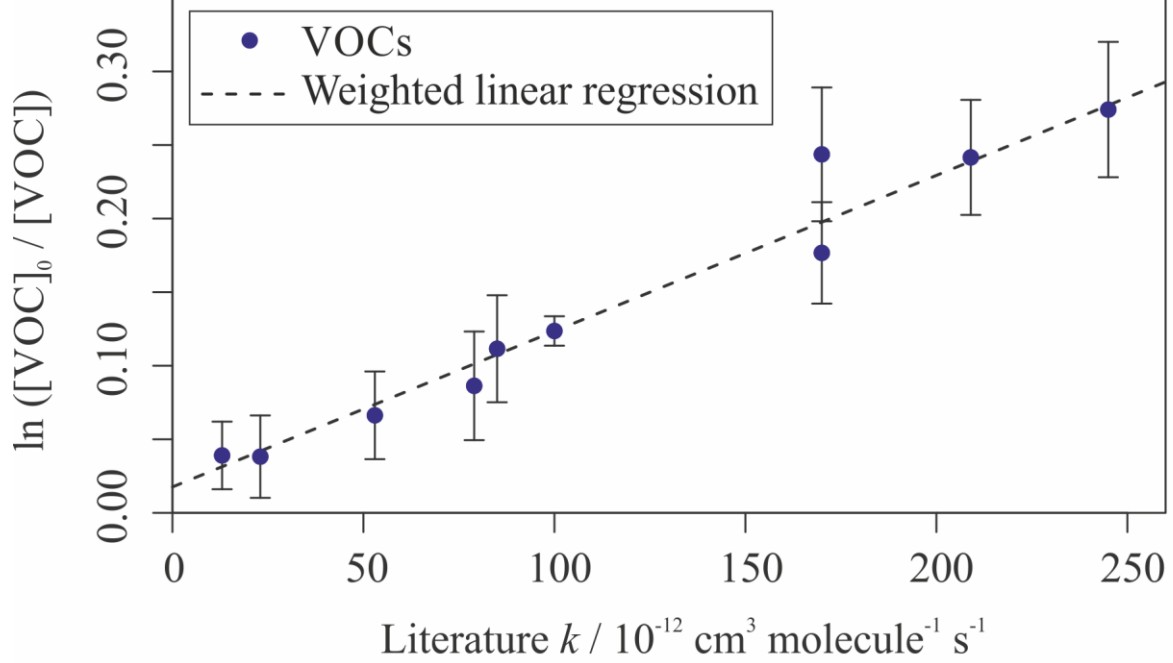

**Figure 3 Relative rate plot for Mixture 1 (OH reactivity = 240 s⁻¹) at 295 K. Compounds with a known rate coefficient are plotted using literature values. Error bars on the y-axis, equal to one standard error, are calculated by combining the standard error in peak areas for 6 lamp off samples and 6 lamp on samples. Error bars on the x-axis are not displayed as they are typically large (approximately ± 20-30%) and account for deviations from the line. A weighted (to the uncertainty in the y-values) linear fit was used to generate the slope with a value of $OH_{exp}$ = 1.1 (± 0.07) × 10⁹ molecules cm⁻³ s and R²of 0.96.**





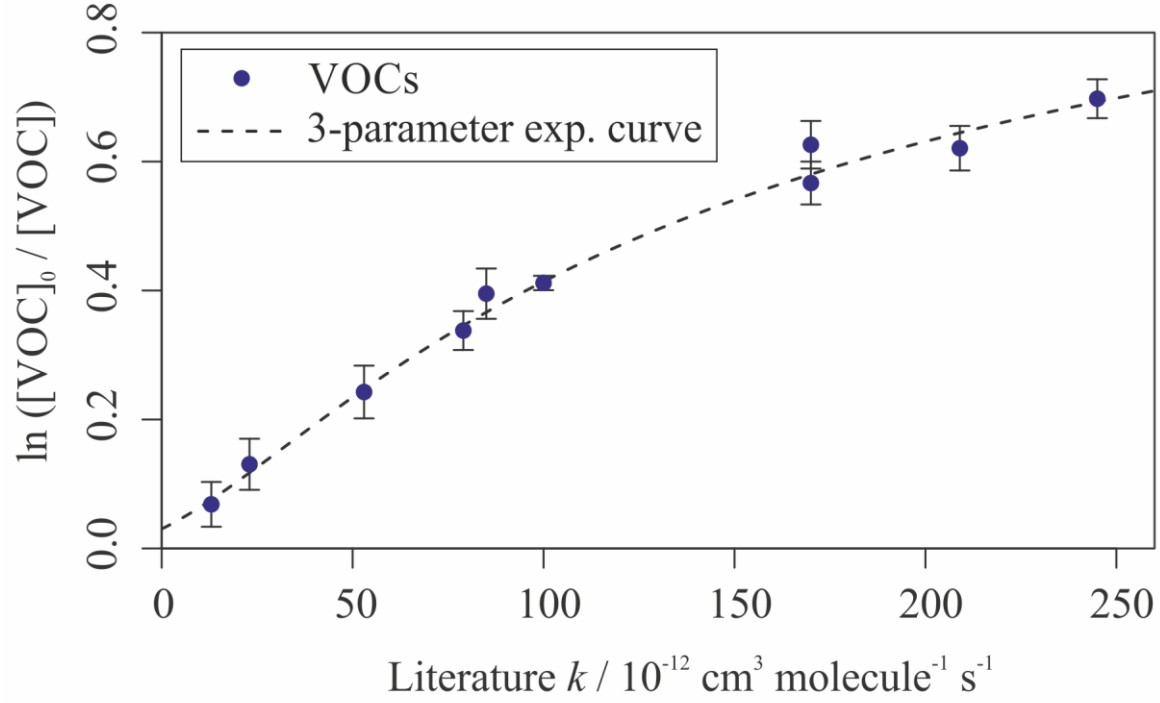

**Figure 4 Relative rate plot for Mixture 1 (OH reactivity = 50 s⁻¹) at 295 K. Compounds with a known rate coefficient are plotted using literature values. Error bars on the y-axis, equal to one standard error, are calculated by combining the standard error in peak areas for 6 lamp off samples and 6 lamp on samples. Error bars on the x-axis are not displayed as they are typically large (approximately ± 20-30%) and account for deviations from the line. The black dashed line shows the relationship observed when VOC concentrations are low, modelled by a three-parameter exponential curve.**



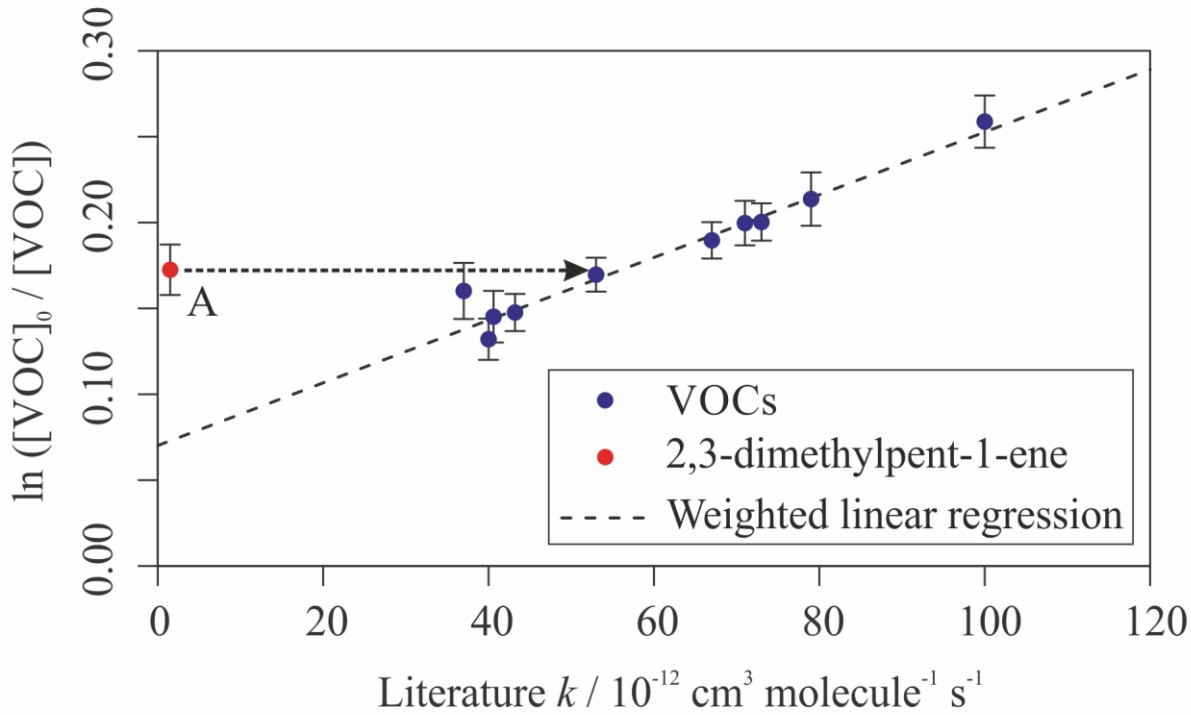

**Figure 5 Relative rate plot for Mixture 2 (OH reactivity = 30 s⁻¹) at 295 K. Compounds with a known rate coefficient are plotted using literature values. Error bars on the y-axis, equal to one standard error, are calculated by combining the standard error in peak areas for 6 lamp off samples and 6 lamp on samples. A weighted linear fit was used to generate the slope with a value of $OH_{exp}$ = 1.8 (± 0.14) × 10⁹ molecules cm⁻³ s and R² of 0.95. Data for 2,3-dimethylpent-1-ene, which has no literature $k$ value, was not used in the calculation of the fit.**



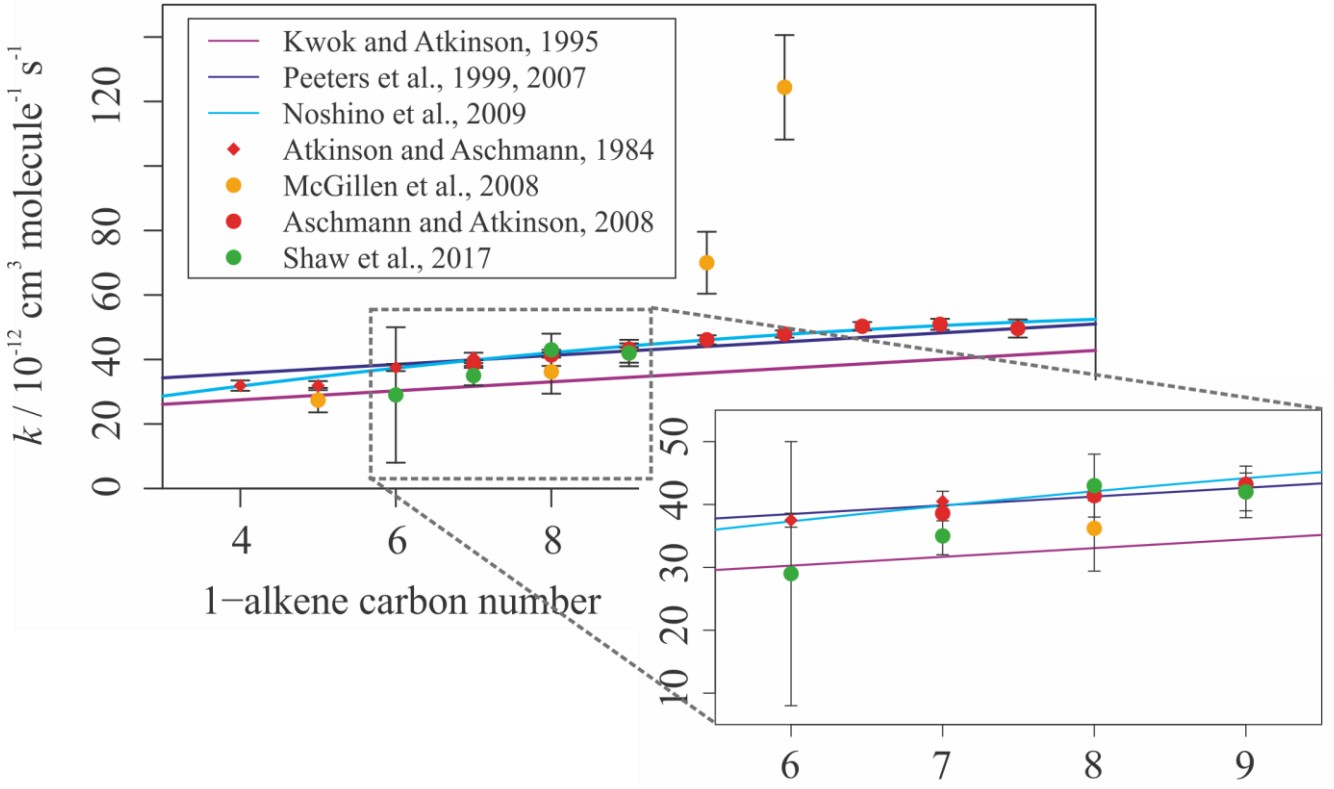

**Figure 6** Plot showing experimentally derived (data points) and predicted SAR derived (lines) *k* values for 1-alkenes. Data from this study are shown in green; they are in good agreement with both previous experimentally derived results and with theoretical SARs.





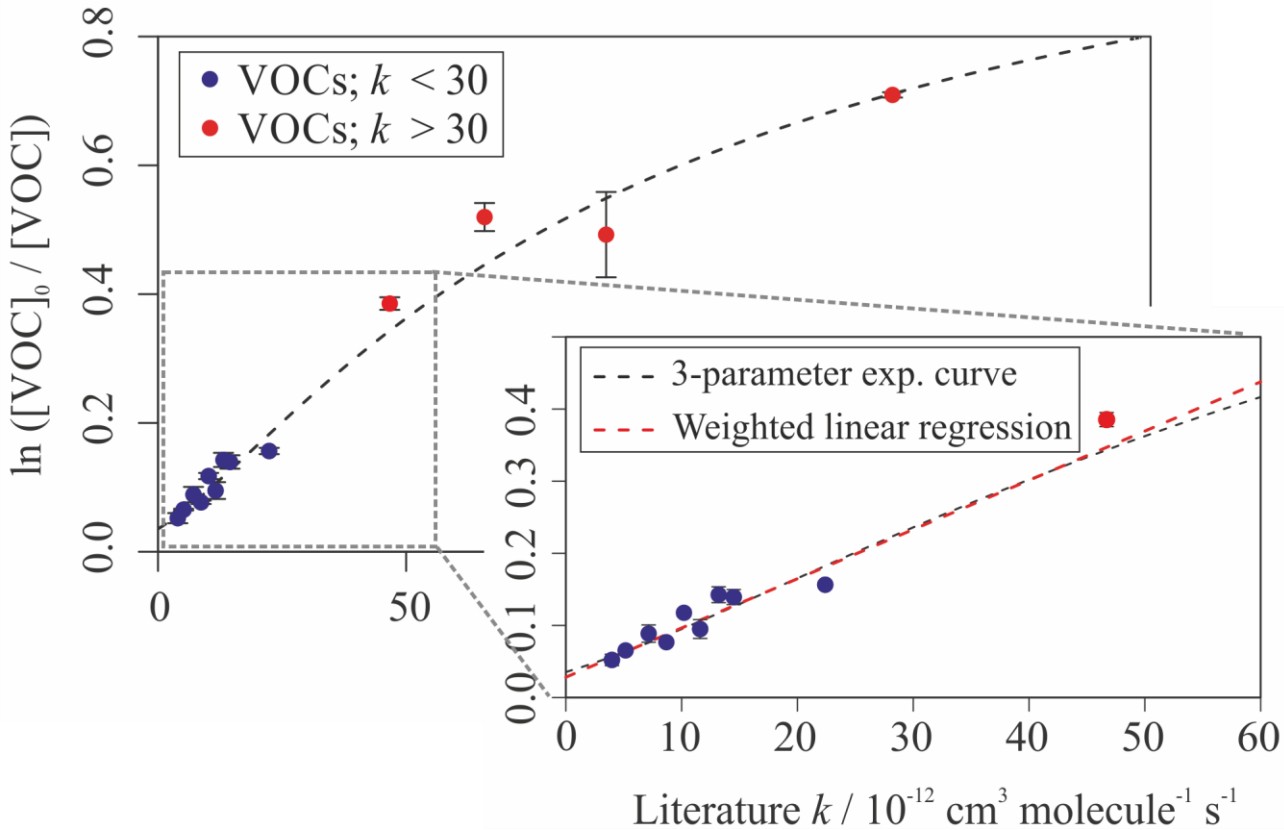

**Figure 7 Relative rate plot for Mixture 3 (OH reactivity = 60 s⁻¹) at 323 K. Compounds with a known rate coefficient are plotted using literature $k$ (323 K) values. Error bars on the y-axis, equal to one standard errors, are calculated by combining the standard error in peak areas for 5 lamp off samples and 5 lamp on samples. The black dashed line shows the relationship observed when VOC concentrations are low, modelled by an exponential cumulative distribution. A weighted linear fit was used to generate the red slope (R² of 0.976) for VOC with $k$ (323 K) less than $30 \times 10^{-12}$ cm³ molecule⁻¹ s⁻¹.**



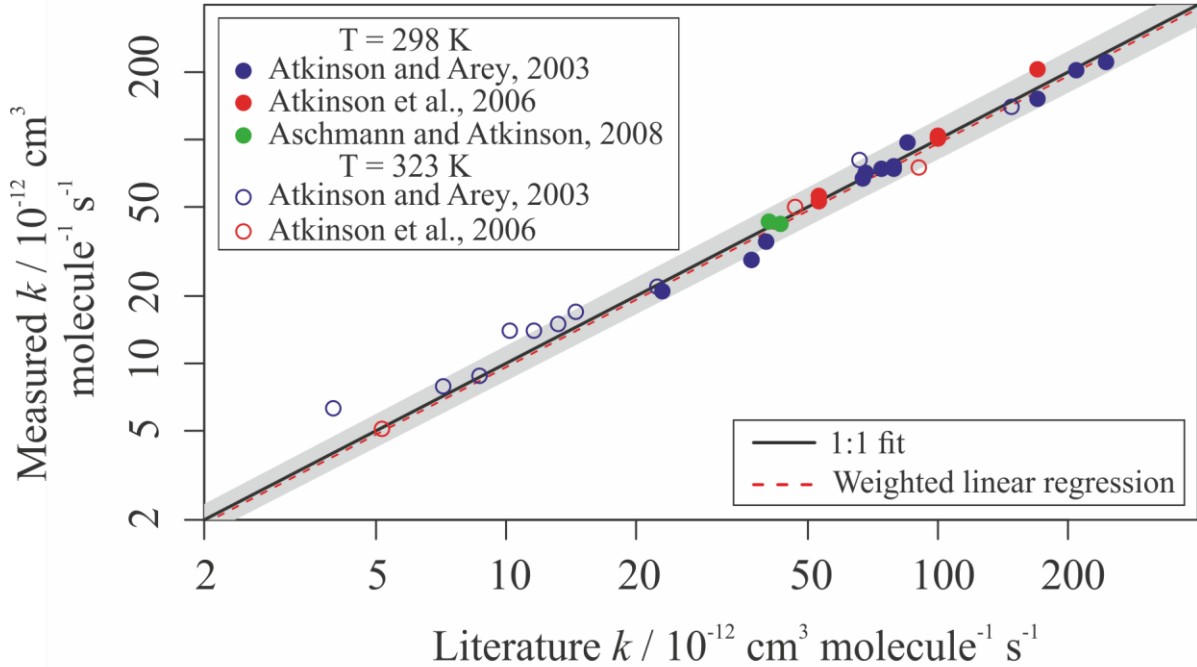

**Figure 8 Measured VOC + OH rate coefficients plotted against literature VOC + OH rate coefficients for all compounds measured as part of mixtures 1, 2 and 3. Filled points represent data collected at room temperature (294 (± 2) K) and empty points represent data collected at the elevated temperature of 323 (± 10) K. The grey shaded area demonstrates a 25% uncertainty in the 1:1 gradient; most data falls well within this bound. Data lying outside this bound include: 1-hexene at room temperature and 2,2,3-trimethylbutane, n-nonane and β-pinene at 323 K. Weighted linear regression analysis for the data yields a slope with equation y = 1.00 (± 0.02) x + 0.96 (± 0.5) and an R$^2$ of 0.98.**





**Figure 9 Schematic of the OH reactor configuration used for sampling ambient air VOC. Key to abbreviations: CIA8 = air server and canister interface accessory; GC = gas chromatograph; MFC = mass flow controller; TOF-MS = time of flight mass spectrometer; TDU = thermal desorption unit.**





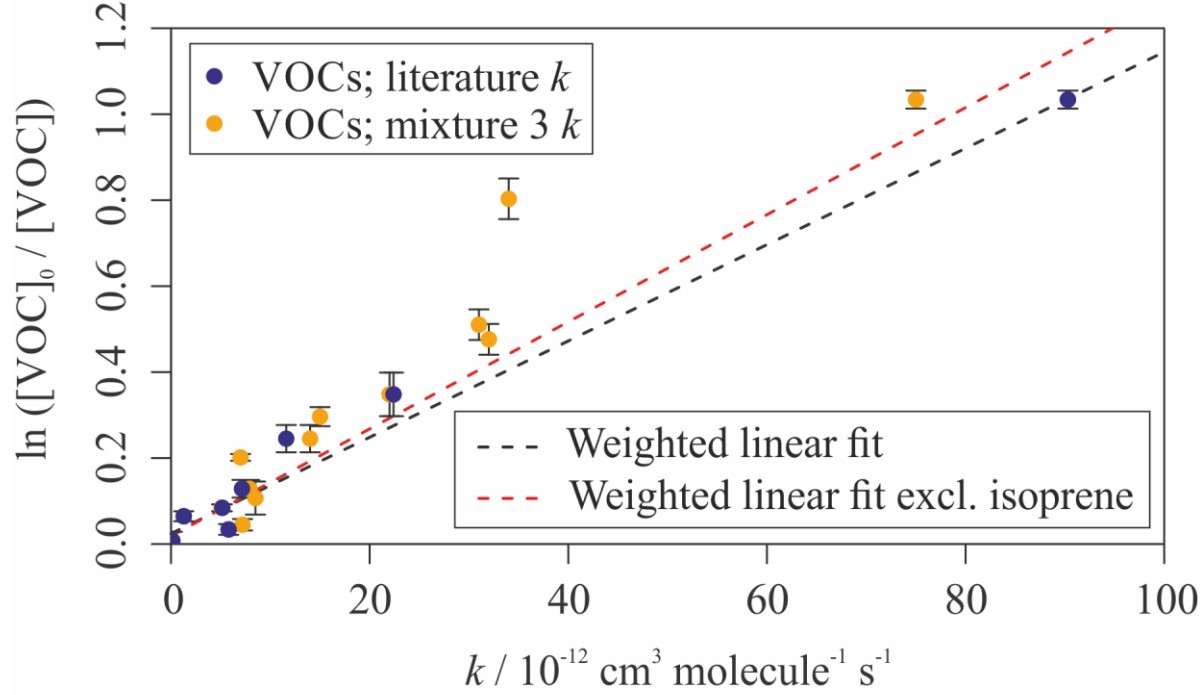

**Figure 10 Relative rate plot for ambient air analysis. Compounds with a known rate coefficient are plotted using literature values at 323 K. A weighted linear fit of this data was used to generate the black dashed slope with an $R^2$ of 0.98. A second weighted linear fit, excluding isoprene, is shown by the red dashed slope ($R^2$ of 0.71). Some of the compounds do not have literature rate coefficients at 323 K but were measured at this temperature elsewhere in this paper (see Table 3). Compounds with a rate coefficient derived from Mixture 3 relative rate studies are also plotted (in yellow) to compare between the synthetic mixture and ambient air results. Error bars on the y-axis, equal to two standard errors, are calculated using the standard deviation in depletions for each VOC across all samples. Error bars on the x-axis are not shown (up to 35%).**

