# Peer review of "A self-consistent, multi-variate method for the determination of gas phase rate coefficients, applied to reactions of atmospheric VOCs and the hydroxyl radical"

_Atmospheric Chemistry and Physics, 2017_

## Referee Comment (RC1) · Anonymous Referee #1 · 23 Oct 2017

This is excellent work. I agree with the authors claim that this marriage of modern high-throughput analytical techniques with the well established and robust relative rate method represents a significant breakthrough in our ability to measure the kinetics of relevant atmospheric reactions. I look forward to this technique being used to substantially expand the atmospheric chemistry kinetic database, particularly at temperatures other than 298K.

---

## Referee Comment (RC2) · Anonymous Referee #2 · 25 Oct 2017

Shaw at al present a modified form of the relative rate method to derive self-consistent rate coefficients for reaction of OH with several VOCs simultaneously. The method is validated by comparison with a suite of OH-rate coefficients for various VOCs as found in compilations and evaluations of the kinetics literature.

The authors suggest that this method represents an improvement on the relative rate technique because of the higher throughput in terms of rate coefficients measured and also because of the multi-variate analysis, which does not rely on a single reference compound. I agree with the first statement, but would argue that the conventional

relative rate method, with a single reference reactant also has advantages in terms of traceability (and thus simple future correction) of the rate coefficients obtained and also in the extension of the linear region over which the underlying kinetic framework works. Although Shaw et al show that the rate coefficients are in good agreement with the literature their treatment of the non-linearity of the data (i.e. plots of ln(depletion factor) v k(literature)) is very weak and demands a more rigorous explanation.

The authors should address the following points in a revised manuscript:

P3L10 "Thus, this aspect of gas-phase kinetics represents a fundamental pillar of atmospheric and, more widely, environmental science." The importance of kinetics, and laboratory studies as a whole, has been recently highlighted by Burkholder et al.(J., Env. Sci. Tech., 51, 2519-2528, 2017) which could be cited here.

P3L17 replace "such as rate coefficient" with "such as the rate coefficient"

P3L18 replace "method" with "semi-empirical method"

P3L29 replace "which ideally react at a similar rate" with "which ideally has a similar rate coefficient"

P3L30 replace "the ratio of their depletion allows" with "the ratio of their depletion factors (or fractional depletion) allows"

P4L25 "Rather than rely on one reference VOC, a subset of the existing OH + VOC kinetic database is used to place the experimental relative rate data on an absolute scale, thereby reducing uncertainties." I do not completely agree. The uncertainty will depend ultimately on the quality of the literature data for several trace gases and depend sensitively on how the real uncertainties in each of these literature rate coefficients are 1) assessed and 2) how they are used (weighting) in the fitting procedure. I do not think that a singly measured rate coefficient compared to one (or two) well known reference reactants in separate experiments is less accurate. It might take more time (per VOC), but it will be more reliable.

P4L30 "near ambient" ?. VOC oxidation in the atmosphere occurs over a wide range of temperatures and pressures. Perhaps "ambient" is the wrong term here as conditions of one bar and 295-323 K represent only a small fraction of the atmosphere.

P5L23 "depletion" Perhaps fractional loss of a VOC ?

P5L31 "despite the establishment of laminar flow". "Because of laminar flow rather than despite laminar flow ? Laminar flow at atmospheric pressure reduces mixing. Turbulent flow forces mixing. Is the flow really laminar ? Give the Reynolds number and entrance length for acquiring laminar flow.

P6L1 "When using reactive gas mixtures containing VOCs which react rapidly with OH, a non-linear relationship between ln ([VOC]0[VOC]) and k may be observed, as compounds experience differing exposures to OH. Slow reacting VOCs may experience a larger exposure to OH relative to faster reacting VOCs." This is the first time that this phenomenon is addressed in the manuscript and the explanation is not helpful. In a mixture of reactive gases, the number of OH reactions (integrated over time) of any single VOC will depend on the (time dependent) steady state OH concentration, which itself depends on the overall reactivity and its production rate. Slow reacting VOCs will be less depleted than fast reacting VOCs, but the relative change should still be defined by the relative rate coefficient.

P6L10 " . . . the use of multiple known k values should minimise any systematic error induced compared with using a single reference compound." This statement is too broad. Systematic error will depend on how well known the reference rate coefficients are and how close they are (in depletion factors) to the unknown VOC. i.e. a reference VOC with a rate coefficient that is a factor 10 faster and which has a large associated uncertainty will not improve the accuracy of the k derived.

P7L1 Perhaps worth mentioning the implicit assumption that trace gases are not selectively lost in the trapping (reaction with condensed water) or desorption process (via thermal decomposition).

P8L4 How was the total OH reactivity estimated if one compound did not have a rate coefficient prior to conducting the experiments ?

P9L14 "However, despite the intercept being close to zero, linear fits with a non-zero intercept were found to be a more appropriate approximation" This is a wishy-washy statement. How is "more appropriate" defined and why ?

P9L15 Replace "needs to only be a..." with "needs only to be a..."

P9L16 "a three-parameter exponential distribution function, given in Eq. (5) was used to fit the data..." How was this function chosen ? Would it not make more sense to select a function that goes through zero depletion when k_literature is zero. This becomes more apparent when you examine Figure 7.

P9L17 "4. Whilst this is inconsistent with Eq. (3), it does not necessarily detract from the relative rate nature of the experiment as a reliable and consistent function can still be plotted through the data. This type of distribution possibly occurs when gas mixtures contain low concentrations of very fast reacting VOCs and arises due to poor mixing conditions within the reactor." I find this aspect of the work needs more attention. The plot is in fact linear up to a depletion of reactants of 35-40 % and in this regard is completely compatible with the data shown in Figure 3. The curvature occurs at large depletion. Would curvature be apparent at a reactivity of 240 /s if there were more time for reaction or more OH available (to increase the depletion factors) ? The authors rather hand-waving explanation about mixing needs to be more rigorously discussed as this might represent a limitation of this method. Note that similar (non-linear) effects were reported by Sinha et al in their paper on the comparative reactivity method, which was found to be dependent on the reactivity of the system and which could be explained (via numerical simulation) as arising from a departure from first-order kinetics. Perhaps simulations could shed some light on these effects here too ?

P9L24 "The gradient of the line in Fig. 3 is equal to the integral of OH concentration over time." This can also be calculated from the OH concentration, derived from the

depletion rate of a single VOC and the flow parameters of the reactor. The authors should compare this. It would in any case be interesting to read what the production rate of primary OH is and what OH levels are present.

P9L26 "Li et al. (2015) and Peng et al. (2015) found that increasing the OH reactivity results in an increased rate of removal of OH radicals from the system." I don't understand this statement. Increasing the OH reactivity must always increase the rate of removal of OH (which, at any point in time, is [OH]*reactivity). Are the authors indicating that changes in the recycling of OH may occur for systems with more highly reactive hydrocarbons or is this a manifestation of secondary reactions of OH with the products formed (which may be more or less reactive that the initial mixture) ?

P9L29 "Using Eq. (3) it is possible to estimate new k values at room temperature (295 ($\pm$ 2) K) for all components in Mixture 1..." Then the equation must be modified as it presently only deals with isoprene reacting with OH. As the depletion factor for isoprene (data point at 100e-12) has the lowest associated uncertainty in Fig 3, the weighted fit is forced to pivot through this data point. I would like to see the uncertainty in the literature k on this plot (and on Figs 4-7) and also read more about the assessment of uncertainty and the weighting procedure used in the fit (which presumably took errors in both the x and y directions into account).

To make the plots more readable, I suggest labelling each data point with the VOC concerned, otherwise one has to refer to the tables with the literature rate coefficients to see which data point is which VOC.

P10L1 "The majority of VOCs............." Here we are dealing with the advantages over the conventional relative rate approach. Many rate coefficients in the literature have been derived using relative rate methods and a certain set of "common" reference reactants chosen because they have well defined rate constants. For any particular pair of reactants, the uncertainty of the rate constant derived by the relative rate method is lowest when reference and reactant have similar rate coefficients. One could

argue that mixing in reactants that react much more slowly (or quickly) actually reduces the accuracy. Also, unlike in the conventional relative rate method, where changes (i.e. improvements over time) in the rate constant for a reference reactant can be transferred easily to correct all values obtained for other gases measured relative to it, the relationship in the multivariate case is more obscure.

P10L8 Replace "measurements" with "rate coefficient measurements"

P10L12 Are the Hites et al measurements relative or absolute ? Delete the comma after "rate"

P10L14 Replace "limonene oxidation rate . . ." with "the rate coefficient for reaction of limonene with OH . . ."

P10L16 Replace "value" with "rate coefficient"

P10L17 "this is an erroneous measurement that arose due to the OH initiated depletion of o-xylene being too small and subject to large relative errors". Not sure about this explanation. A seven percent change should be sufficient to get within about 20-30 % of the true number. A factor 10 disagreement must have a different origin.

P10L19 "Figure 5 shows an example relative rate plot for Mixture 2; for which a linear relationship is observed across all OH reactivities tested". As for Figures 3 (non-zero intercept) and 4 (very obvious curvature at large depletion factors), the data do not lie on a straight line but on a curve. This is the reason for the large intercept. The data cover depletions of between 13 % and 25 % only and a straight line fit will of course go through the data points in this small range. The question that needs to be addressed here is why the intercept (at a rate constant of zero) indicates large depletion still.

P11L1 "order to prevent excessive partitioning of the less-volatile. . . .." Be more precise. what is excessive in this context ? what might be the effect of partitioning to the walls in terms of its impact on the relative concentration changes after reaction ?

P11L4 "Likely due to the small concentrations of VOCs injected and the large range

in VOC + OH rate coefficients" is not an explanation of the phenomenon. The non-linearity in these plots requires more rigorous examination.

P11L9 The value obtained for myrcene is actually (within combined uncertainty) in agreement with the literature value listed.

P11L19 report the gradient along with its standard deviation.

P11L26 "Whilst there is still a good correlation between ..." Does the "good correlation" refer to the fact that the "non-linearity" can be removed by a fitting function ? In this case, any data that can be fitted with a function (no matter how complex) could be described as well correlated. This tends to disguise the fact that the relationship between the depletion factors and k_literature is not entirely understood.

P12L30 OH + aromatics. To what extent do these reactions involve reversible addition of OH to the aromatic ring, which can then react with O2 (or not in this case as the experiments are in N2). Can the lack of O2 alter the effective (forward) rate coefficient ?

P13L3 "and can only include compounds for which authentic standards or pure raw materials are available." Why pure ? An impure sample will work perfectly well as long as it can be selectively detected. This is the main advantage of relative rate versus absolute methods.

P13L14 "There is also likely to be a large variety in the concentrations at which atmospheric VOCs exist". I think this is proven rather than "likely".

P13L15 "which could have an adverse effect on the uniformity of individual VOC exposure to OH" What is the "adverse effect" and what does "uniformity of exposure to OH" mean ?

P14L15 "OH. However, a significant section of the plot lacks literature data for comparison" In this case it would be beneficial to spike the air with a few reference gases with well-known rate coefficients before entering the reactor. P15L13 "this method can pro-

duce novel results. . ." perhaps "can produce rate coefficients for VOCs, the OH-kinetics of which have not been investigated to date" or similar.

Figures

Figures 3-5 and 7 should be improved by 1) adding the error bars representing the uncertainty in k. I consider this to be very important in order to assess the quality of the data. 2) identifying the VOCs.

Figure 7: The inset obscures a large part of the x-axis of the main plot. This can be done better.

---

## Referee Comment (RC3) · Anonymous Referee #3 · 29 Nov 2017

This manuscript describes flow tube studies in which OH radicals are reacted with a mixture of volatile organic compounds (N>10) simultaneously, in order to generate a self-consistent set of reaction rate coefficients. The concept for the experiment is good, but I question the generality, and hence utility, of the method.

The experiments are in general well-described. The manuscript contains clear figures, and also details about dimensions and flow rates used, which always help in doing quick calculations of residence times, etc. The manuscript itself contains some errors and inconsistencies, which will be detailed below. Of course, the simultaneous measurement of rate coefficients by the relative rate method is not new. Kramp and Paulson [J. Phys. Chem. A, 102, 1998, 2685-2690] reacted mixtures of hydrocarbons in Teflon bags to evaluate systematic uncertainties between reference compounds. However, to the best of my knowledge, this has not been done in a flow reactor until now.

The biggest issue with the manuscript is that the method is not actually universal, since the derived rate coefficient is dependent on the reactivity of the mixture and the extent of conversion. Thus, it appears that knowledge of the rate coefficients is almost required in order to put them on an absolute basis. For this reason, I find that the methodology would have limited applicability to other systems. For example (and this is touched on in the manuscript) in ambient air samples the reactivity can vary widely, and so one needs to know whether the response will be linear or not. Also, if someone else were going to set up a similar experiment, would they have to characterize their system with curves such as those shown in Figure 4? The manuscript does not really answer these questions and reads a little like a work in progress for me, at least.

For these reasons I can not recommend publication of the manuscript in its current form. Minor corrections follow (format Page/Line).

Page 4/14. The subject of this sentence is "this parameter" (i.e. the reactivity) so it should not vary from milliseconds to tens of seconds.

Page 4/18. I would imagine that lack of knowledge of the identity of VOCs is more responsible for missing reactivity than errors in the rate coefficients? Experimental section. Sometimes cc are used, sometimes ml for volume. Please be consistent.

Page 8/3. Remove "a" before "magnitude".

Page 8/25, and caption for Figure 2. It says here and in the caption that black is lamp off, blue is lamp on, while in the following sentence and the Figure legend this is reversed (and presumably correct).

Page 9/12, and caption for Figure 3. OH reactivity is given as 180 in text, 240 in Figure

caption.

Page 12, paragraph starting on line 19. This is a little confusing. By non-Arrhenius, I presume you mean having a negative activation energy. This would be expected for addition reactions. However, the last sentence does not really make sense. I do not know if the differences you see are meaningful at this level of precision.

Page 12/27. I would think that all OH + alkane rate coefficients would show a "normal" activation energy, linear, cyclic or branched.

Fig 6 Noshino should be Nishino

McGillen (2008) is missing from ref list

---

## Author Comment (AC1) · 5 Feb 2018

**Authors' response to all referees for ACP-2017-917**

The authors would like to thank all the referees for their helpful and insightful comments. Below is a breakdown of all referee comments (*black italics*) with appropriate author responses (blue text).

**Anonymous Referee #1**

*This is excellent work. I agree with the authors claim that this marriage of modern high-throughput analytical techniques with the well established and robust relative rate method represents a significant breakthrough in our ability to measure the kinetics of relevant atmospheric reactions. I look forward to this technique being used to substantially expand the atmospheric chemistry kinetic database, particularly at temperatures other than 298K.*

    The authors would like to thank the referee for their supportive review.

**Anonymous Referee #2**

*Shaw at al present a modified form of the relative rate method to derive self-consistent rate coefficients for reaction of OH with several VOCs simultaneously. The method is validated by comparison with a suite of OH-rate coefficients for various VOCs as found in compilations and evaluations of the kinetics literature.*

*The authors suggest that this method represents an improvement on the relative rate technique because of the higher throughput in terms of rate coefficients measured and also because of the multi-variate analysis, which does not rely on a single reference compound. I agree with the first statement, but would argue that the conventional relative rate method, with a single reference reactant also has advantages in terms of traceability (and thus simple future correction) of the rate coefficients obtained and also in the extension of the linear region over which the underlying kinetic framework works.*

    The authors agree with the referee's comment that using a single reference reactant has advantages in traceability.

    The manuscript has been updated to clarify this and now reads (P10L10): "*By using multiple reference compounds, the risk of a single erroneous reference value perturbing the rest of the data is reduced, assuming that all compounds behave in a similar way upon exposure to OH. This advantage does come at the expense of a loss in traceability. When changes are made to the reference rate coefficient for a specific measurement it is relatively easy to propagate the uncertainty change through to the relative rate measurements. When using multiple reference reactions, and generating a relationship between them, the transfer of an updated rate coefficient change would obviously be more complicated.*"

*Although Shaw et al show that the rate coefficients are in good agreement with the literature their treatment of the non-linearity of the data (i.e. plots of ln(depletion factor) v k(literature)) is very weak and demands a more rigorous explanation.*

    A number of the referee's comments concern the non-linearity of the data. Comments referring to this phenomenon are listed below, with the corresponding authors response in full below them. Comments referring to other aspects of the manuscript are addressed from page 5 onwards.

*P6L1 "When using reactive gas mixtures containing VOCs which react rapidly with OH, a non-linear relationship between ln ([VOC]0[VOC]) and k may be observed, as compounds experience differing exposures to OH. Slow reacting VOCs may experience a larger exposure to OH relative to faster reacting VOCs." This is the first time that this phenomenon is addressed in the manuscript and the explanation is not helpful. In a mixture of reactive*

*gases, the number of OH reactions (integrated over time) of any single VOC will depend on the (time dependent) steady state OH concentration, which itself depends on the overall reactivity and its production rate. Slow reacting VOCs will be less depleted than fast reacting VOCs, but the relative change should still be defined by the relative rate coefficient.*

*P9L17 "4. Whilst this is inconsistent with Eq. (3), it does not necessarily detract from the relative rate nature of the experiment as a reliable and consistent function can still be plotted through the data. This type of distribution possibly occurs when gas mixtures contain low concentrations of very fast reacting VOCs and arises due to poor mixing conditions within the reactor." I find this aspect of the work needs more attention. The plot is in fact linear up to a depletion of reactants of 35-40 % and in this regard is completely compatible with the data shown in Figure 3. The curvature occurs at large depletion. Would curvature be apparent at a reactivity of 240 /s if there were more time for reaction or more OH available (to increase the depletion factors) ? The authors rather hand-waving explanation about mixing needs to be more rigorously discussed as this might represent a limitation of this method. Note that similar (non-linear) effects were reported by Sinha et al in their paper on the comparative reactivity method, which was found to be dependent on the reactivity of the system and which could be explained (via numerical simulation) as arising from a departure from first-order kinetics. Perhaps simulations could shed some light on these effects here too ?*

*P10L19 "Figure 5 shows an example relative rate plot for Mixture 2; for which a linear relationship is observed across all OH reactivities tested". As for Figures 3 (non-zero intercept) and 4 (very obvious curvature at large depletion factors), the data do not lie on a straight line but on a curve. This is the reason for the large intercept. The data cover depletions of between 13 % and 25 % only and a straight line fit will of course go through the data points in this small range. The question that needs to be addressed here is why the intercept (at a rate constant of zero) indicates large depletion still.*

*P11L4 "Likely due to the small concentrations of VOCs injected and the large range in VOC + OH rate coefficients" is not an explanation of the phenomenon. The nonlinearity in these plots requires more rigorous examination.*

The referee is correct, significant curvature occurs at depletions larger than 40 %, with most plots being linear up to a depletion factor of approximately 0.4. Up to this point, rate coefficient estimations made using either the curved relationship or linear regression are in good agreement, within errors (see Table 3). It is currently not possible to experimentally determine what the impact of a larger initial [OH] would be at higher OH reactivity.

Model simulations have been performed to try and understand the reason behind the observed deviation from linearity. The simulations were conducted using Kintecus and incorporated the OH + VOC reactions for Mixture 1, as well as very simple $HO_x$ chemistry (OH + OH $\rightarrow$ $H_2O$ + O, $k$ = $1.48 \times 10^{-12}$ $cm^3$ molecule$^{-1}$ s$^{-1}$; OH + OH $\rightarrow$ $H_2O_2$, $k$ = $6.20 \times 10^{-12}$ $cm^3$ molecule$^{-1}$ s$^{-1}$; OH + $HO_2$ $\rightarrow$ $H_2O$ + $O_2$, $k$ = $1.10 \times 10^{-10}$ $cm^3$ molecule$^{-1}$ s$^{-1}$; $HO_2$ + $HO_2$ $\rightarrow$ $H_2O_2$ + $O_2$, $k$ = $1.60 \times 10^{-12}$ $cm^3$ molecule$^{-1}$ s$^{-1}$).

To simulate incomplete mixing of OH, the reactor was divided into three distinct theoretical air parcels; one where ⅓ of the [VOC] is exposed to a low [OH], one where ⅓ [VOC] is subjected to a high [OH] and one where ⅓ [VOC] is exposed to a level of [OH] between these two high and low values. The depletion factors of the VOCs in each of these air parcels can be plotted separately, or summed together to yield an average depletion factor. The average depletion factor is more representative of what is actually measured by the GC-MS, assuming that thorough mixing of all air parcels takes place after the reactions with OH have occurred.

Figure S1 shows that in each of these air parcels, the VOCs are depleted relative to their rate coefficients. The adjusted $R^2$ values for each of the three linear regressions is 1.0.

[Figure]

Figure S1. Simulated depletion factors for Mixture 1 (OH reactivity = 50 s⁻¹) plotted against literature rate coefficients. Here three air parcels are simulated separately, with each air parcel containing ⅓ [VOC] and exposed to different [OH] (molecules cm⁻³).

However, if the concentrations in each section of the reactor are summed before the depletion factor is calculated, as happens prior to sampling by the GC-MS, the resulting plot against rate coefficient is indeed curved, as shown in Fig. S2.

[Figure]

Figure S2. Simulated depletion factors for Mixture 1 (OH reactivity = 50 s[-1]) plotted against literature rate coefficients. Here the concentrations of the VOCs in each of the three simulated air parcels (Fig. S1) are summed and the depletion factor for the final concentration is plotted against literature rate coefficient.

This curvature still occurs at higher [VOC], but is much less pronounced, to the extent that the relationship can be assumed to be linear. Figure S3 shows the same simulation for a [VOC] of 10 ppb, or OH reactivity of 290 s[-1]. Note that the model reproduces the discrepancy in depletion factor observed experimentally and that the curvature is only really noticeable when the depletion factor is greater than 0.4.

[Figure]

Figure S3. Simulated depletion factors for Mixture 1 (OH reactivity = 290 s[-1]) plotted against literature rate coefficients. Here the concentrations of the VOCs in each of the three simulated air parcels are summed and the depletion factor for the final concentration is plotted against literature rate coefficient.

Whilst this artefact of the experiment does represent a limitation, at least in terms of sensitivity towards measuring rate coefficients towards the greater end of the reactivity spectrum, it does not affect measurements made for VOCs with a depletion factor less than 0.4. As mentioned before, if a consistent function can be plotted through the data and an unknown is interpolated using that function, the relative rate conditions should still apply.

This model analysis, explaining the observed curvature, has been added to the Supplementary Information in a section titled 'Kinetics simulations of poor mixing resulting in non-linearity'.

Authors' response to all referees for ACP-2017-917

*The authors should address the following points in a revised manuscript:*

*P3L10 "Thus, this aspect of gas-phase kinetics represents a fundamental pillar of atmospheric and, more widely, environmental science." The importance of kinetics, and laboratory studies as a whole, has been recently highlighted by Burkholder et al.(J., Env. Sci. Tech., 51, 2519-2528, 2017) which could be cited here.*

> Citation added.

*P3L17 replace "such as rate coefficient" with "such as the rate coefficient"*

> Amended to "*parameters such as the rate coefficient…*".

*P3L18 replace "method" with "semi-empirical method"*

> Amended to "*thereby providing a semi-empirical method…*"

*P3L29 replace "which ideally react at a similar rate" with "which ideally has a similar rate coefficient"*

> Amended to "*which ideally possess similar rate coefficients…*".

*P3L30 replace "the ratio of their depletion allows" with "the ratio of their depletion factors (or fractional depletion) allows"*

> Please refer to comment P5L23.

*P4L25 "Rather than rely on one reference VOC, a subset of the existing OH + VOC kinetic database is used to place the experimental relative rate data on an absolute scale, thereby reducing uncertainties." I do not completely agree. The uncertainty will depend ultimately on the quality of the literature data for several trace gases and depend sensitively on how the real uncertainties in each of these literature rate coefficients are 1) assessed and 2) how they are used (weighting) in the fitting procedure. I do not think that a singly measured rate coefficient compared to one (or two) well known reference reactants in separate experiments is less accurate. It might take more time (per VOC), but it will be more reliable.*

> The authors agree that the use of multiple reference reactions does not reduce uncertainties but rather reduces the reliance on single reference reactions. The manuscript now reads "*A subset of the existing OH + VOC kinetic database is used to place the experimental relative rate data on an absolute scale, thereby reducing the reliance on a single reference reaction.*"

*P4L30 "near ambient" ?. VOC oxidation in the atmosphere occurs over a wide range of temperatures and pressures. Perhaps "ambient" is the wrong term here as conditions of one bar and 295-323 K represent only a small fraction of the atmosphere.*

> Amended to "*Experiments were conducted under conditions of 1 bar (N$_2$) and 298-323 K…*" to clarify the reactor conditions.

*P5L23 "depletion" Perhaps fractional loss of a VOC ?*

> 'Depletion' is not an adequate term to describe the parameter here. Any equivalent mentions of 'depletion' within the manuscript have therefore been amended to 'depletion factor' to aid in clarification. Depletion factor refers specifically to the parameter $\ln\left(\frac{[\text{VOC}]_0}{[\text{VOC}]}\right)$, described on P5L24 in Eq. (3), and not to a depletion in terms of concentration.

*P5L31 "despite the establishment of laminar flow". "Because of laminar flow rather than despite laminar flow ? Laminar flow at atmospheric pressure reduces mixing. Turbulent flow forces mixing. Is the flow really laminar ? Give the Reynolds number and entrance length for acquiring laminar flow.*

To clarify with regards to the Reynolds number ($R_e$): the initial flow of humidified $N_2$ (2000 sccm) has Re = 88 with an entrance length of 0.1 m. The flow after injection of the VOC mixture (total of 3000 sccm) has Re = 132 and an entrance length of 0.2 m. The low values for Re suggests that laminar flow is established within the reactor, both initially and after injection of the VOCs. However, these calculations do not take into account that the VOC mixture is injected perpendicular to the main flow, which is likely to cause considerable turbulence and facilitate mixing.

The manuscript text has been amended to read "*However, because of the establishment of laminar flow in the reactor, thorough and complete mixing of OH is unlikely to take place before the reactions with VOCs occur (see Supplementary Information).*" A section 'On the establishment of laminar flow' has been added to the Supplementary Information to support the text in the manuscript.

*P6L10 " . . . the use of multiple known k values should minimise any systematic error induced compared with using a single reference compound." This statement is too broad. Systematic error will depend on how well known the reference rate coefficients are and how close they are (in depletion factors) to the unknown VOC. i.e. a reference VOC with a rate coefficient that is a factor 10 faster and which has a large associated uncertainty will not improve the accuracy of the k derived.*

The authors agree that this must be clarified further. The text has been amended to "*the use of multiple known k values should reduce the reliance on any single reference reaction*".

*P7L1 Perhaps worth mentioning the implicit assumption that trace gases are not selectively lost in the trapping (reaction with condensed water) or desorption process (via thermal decomposition).*

The authors respectfully disagree that this is a necessary implicit assumption for the GC sampling in this work. Due to this method relying on the relative change in the concentration of VOCs from "lamp off" to "lamp on" samples, so long as the sampling is consistent for the two situations, it should not matter. If a trace gas *is* selectively lost during sampling this will not affect the results if the loss is consistent from "lamp off" to "lamp on".

The text (P5L22) has been amended to "*Assuming consistent trapping and sampling with both the lamp switched off and the lamp switched on, the depletion of an individual VOC, with a known OH rate coefficient can be evaluated using simple kinetic equations.*" to aid in clarification of this.

*P8L4 How was the total OH reactivity estimated if one compound did not have a rate coefficient prior to conducting the experiments ?*

The total OH reactivity is not a parameter necessary for the calculation of *k* values in the paper. The authors thought it useful to include in order to distinguish between differences in mixtures, especially to differentiate between a single mixture that was studied under different flow regimes.

To make this clear, the text (P7L23) has been amended to include the sentence "*This parameter is not necessary for the measurement of k values but is provided in order to differentiate between separate mixtures and also between the same mixture studied under different flow regimes.*"

Regardless, the OH reactivity of a mixture containing an unknown can be estimated in a number of ways, for example, by using a SAR in the absence of experimental *k* values.

Authors' response to all referees for ACP-2017-917

*P9L14 "However, despite the intercept being close to zero, linear fits with a non-zero intercept were found to be a more appropriate approximation" This is a wishy-washy statement. How is "more appropriate" defined and why ?*

'More appropriate' refers to the fit being more precise within the range of OH + VOC *k* values, as we are not concerned with extrapolating to 0 – or indeed outside the range of OH + VOC *k* values. Whilst this is referred to at P6L12, the text at P9L17 has been clarified to read "*However, despite the intercept being close to zero, linear fits with a non-zero intercept were found to be a more appropriate approximation across the range of k values studied. Extrapolation beyond the range of reference OH + VOC* k *values is not necessary to make accurate measurements of* k *values.*"

Adjustments to Figures 3-5 have been made to reflect this – the linear fit has been restricted to only the range of OH + VOC reactions measured, highlighting that knowledge of the intercept is not necessary.

*P9L15 Replace "needs to only be a. . ." with "needs only to be a. . ."*

Amended to "*there needs only to be a…*".

*P9L16 "a three-parameter exponential distribution function, given in Eq. (5) was used to fit the data. . ." How was this function chosen ? Would it not make more sense to select a function that goes through zero depletion when k_literature is zero. This becomes more apparent when you examine Figure 7.*

This function was the only function which consistently and reliably produced a fit to the data. Many other functions could not be resolved using the Origin software for one or more sets of data. The authors agree that the chosen function is not perfect but in the absence of any other, necessitates its use. The authors would like to reiterate their opinion that there needs only be a consistent function that can model the observed data well for the relative rate concept to hold true.

*P9L24 "The gradient of the line in Fig. 3 is equal to the integral of OH concentration over time." This can also be calculated from the OH concentration, derived from the depletion rate of a single VOC and the flow parameters of the reactor. The authors should compare this. It would in any case be interesting to read what the production rate of primary OH is and what OH levels are present.*

An estimate of the [OH] in the reactor may be made by totalling the losses in each VOC, assuming their initial concentrations have been estimated to a reasonable degree. This is likely to produce an underestimation of the [OH] generated by photolysis (due to anticipated losses of OH to $HO_2$, a by-product of $H_2O$ photolysis) but may indicate [OH] availability immediately prior to the injection of VOCs. For Mixture 1, summing the losses in the VOCs, results in an average [OH] = 1.6 ($\pm$ 0.5) $\times 10^{11}$ molecules $cm^{-3}$ over the different OH reactivities tested. For the Mixture 2, the average [OH] = 0.9 ($\pm$ 0.2) $\times 10^{11}$ molecules $cm^{-3}$.

Further insight may be gained using a kinetic model to estimate the extent of OH losses to $HO_2$. Using depletion as a diagnostic, the kinetic model is best matched to experimental results using [OH] = 3.1 ($\pm$ 1.0) $\times 10^{11}$ molecules $cm^{-3}$ for the monoterpenes mixture and [OH] = 1.3 ($\pm$ 0.2) $\times 10^{11}$ molecules $cm^{-3}$ for the alkenes mixture.

These values are of a similar order of magnitude with that calculated for a similar system, in which the reaction of OH with methanol was used to estimate the [OH] produced in the reactor, via the concentration of formaldehyde detected. This was measured to be approximately 2.6 ($\pm$ 1.5) $\times 10^{10}$ molecule $cm^{-3}$ (Cryer, 2016).

*P9L26 "Li et al. (2015) and Peng et al. (2015) found that increasing the OH reactivity results in an increased rate of removal of OH radicals from the system." I don't understand this statement. Increasing the OH reactivity*

*must always increase the rate of removal of OH (which, at any point in time, is [OH]\*reactivity). Are the authors indicating that changes in the recycling of OH may occur for systems with more highly reactive hydrocarbons or is this a manifestation of secondary reactions of OH with the products formed (which may be more or less reactive that the initial mixture) ?*

This section is confusing due to the cyclical nature of arguments. The explanation of OH exposure decreasing with increasing OH reactivity isn't necessary and was therefore removed from the manuscript.

*P9L29 "Using Eq. (3) it is possible to estimate new k values at room temperature (295 (± 2) K) for all components in Mixture 1. . ." Then the equation must be modified as it presently only deals with isoprene reacting with OH. As the depletion factor for isoprene (data point at 100e-12) has the lowest associated uncertainty in Fig 3, the weighted fit is forced to pivot through this data point. I would like to see the uncertainty in the literature k on this plot (and on Figs 4-7) and also read more about the assessment of uncertainty and the weighting procedure used in the fit (which presumably took errors in both the x and y directions into account).*

The authors agree that the referral to Eq. (3) here, which specifically mentions isoprene, is confusing. In order to clarify that a general expression should be used, which is similar in form to Eq. (3) and derived from the weighted linear regression (i.e. taking into account the non-zero intercept), the text was amended to "*By referring to a generalised version of Eq. (3), and using the function of the weighted linear regression, it is possible to estimate new* k *values at room temperature…*".

*To make the plots more readable, I suggest labelling each data point with the VOC concerned, otherwise one has to refer to the tables with the literature rate coefficients to see which data point is which VOC.*

Please refer to the comments on Figs. 3-5 and 7.

*P10L1 "The majority of VOCs. . .. . .. . .. . .." Here we are dealing with the advantages over the conventional relative rate approach. Many rate coefficients in the literature have been derived using relative rate methods and a certain set of "common" reference reactants chosen because they have well defined rate constants. For any particular pair of reactants, the uncertainty of the rate constant derived by the relative rate method is lowest when reference and reactant have similar rate coefficients. One could argue that mixing in reactants that react much more slowly (or quickly) actually reduces the accuracy. Also, unlike in the conventional relative rate method, where changes (i.e. improvements over time) in the rate constant for a reference reactant can be transferred easily to correct all values obtained for other gases measured relative to it, the relationship in the multivariate case is more obscure.*

The use of multiple rate coefficients does have drawbacks in the traceability of measurements, should adjustments to recommended values be made. However, the authors would argue that using a range of multiple reference VOCs, some which react more slowly and some more quickly, gives an advantage in terms of reliability, in that the measurements no longer rely on a single value.

The manuscript has been updated to clarify this and now reads (P10L10): "*By using multiple reference compounds, the risk of a single erroneous reference value perturbing the rest of the data is reduced, assuming that all compounds behave in a similar way upon exposure to OH. This advantage does come at the expense of a loss in traceability; when changes are made to the reference rate coefficient for a specific measurement it is relatively easy to propagate the uncertainty change through to the relative rate measurements. When using multiple reference reactions, and generating a relationship between them, the transfer of an updated rate coefficient change would obviously be more complicated.*"

*P10L8 Replace "measurements" with "rate coefficient measurements"*

Amended to *"the wider literature contains rate coefficient measurements…"*.

*P10L12 Are the Hites et al measurements relative or absolute ? Delete the comma after "rate"*

Amended and included the word 'relative'. The text now reads *"Recent relative rate measurements of the OH + myrcene reaction by Hites and Turner (2009) and Kim et al. (2011)…"*.

*P10L14 Replace "limonene oxidation rate . . ." with "the rate coefficient for reaction of limonene with OH . . ."*

Amended to *"Our measurement of the rate coefficient for the reaction of limonene with OH…"*.

*P10L16 Replace "value" with "rate coefficient"*

Amended to *"The measured rate coefficient for the OH + o-xylene reaction…"*.

*P10L17 "this is an erroneous measurement that arose due to the OH initiated depletion of o-xylene being too small and subject to large relative errors". Not sure about this explanation. A seven percent change should be sufficient to get within about 20-30 % of the true number. A factor 10 disagreement must have a different origin.*

An anomaly in the calculation of error propagation resulted in this discrepancy. This has since been corrected and the value for *o*-xylene adjusted to 4.8 (± 7) × 10$^{-12}$ cm$^3$ molecule$^{-1}$ s$^{-1}$. This is more in line, within error, with the recommended rate coefficient of 13 (± 3) × 10$^{-12}$ cm$^3$ molecule$^{-1}$ s$^{-1}$. However, the large errors on the measured value are still likely due to *o*-xylene lying at the extremes of the reactivity of Mixture 1. Minor adjustments were also made to some other rate coefficients in Mixtures 1 and 2 owing to the same anomaly (12 of the 21 measurements were changed. Of these 12, 9 were adjusted by less than 3 % with *o*-xylene and 1-hexene having adjustments of 67 and 37 % respectively. Note that both these compounds were at the lower extremes of their respective mixtures.)

The manuscript has been amended to *"The measured rate coefficient for the OH + o-xylene reaction, of 4.8 (± 7) × 10$^{-12}$ cm$^3$ molecule$^{-1}$ s$^{-1}$ is the only result which is not totally consistent with the literature. It is likely that this anomaly arose due to o-xylene being at the lower extreme of the mixture reactivity."*

*P11L1 "order to prevent excessive partitioning of the less-volatile. . …" Be more precise. what is excessive in this context ? what might be the effect of partitioning to the walls in terms of its impact on the relative concentration changes after reaction ?*

The wording here is ambiguous: 'excessive' in this context refers to losses which would lower the concentration below the limit of detection for the analytical method. Partitioning to the walls, so long as consistent between lamp off and lamp on samples, should not impact the relative concentration changes. The manuscript was amended to *"in order to avoid greater uncertainties as a result of partitioning of the less-volatile VOCs to the reactor surfaces."*

*P11L9 The value obtained for myrcene is actually (within combined uncertainty) in agreement with the literature value listed.*

This is correct and the manuscript has been updated accordingly: "One measurement stands out as contrasting with its literature value; that for β-ocimene, of 950 (± 800) × 10$^{-12}$ cm$^3$ molecule$^{-1}$ s$^{-1}$."

*P11L19 report the gradient along with its standard deviation.*

Amended to *"The gradient of this regression is 1.0 (± 0.02)…"*.

*P11L26 "Whilst there is still a good correlation between . . ." Does the "good correlation" refer to the fact that the "non-linearity" can be removed by a fitting function ? In this case, any data that can be fitted with a function (no matter how complex) could be described as well correlated. This tends to disguise the fact that the relationship between the depletion factors and k_literature is not entirely understood.*

Amended to "*Whilst there is still a correlation between $\ln\left(\frac{[VOC]_0}{[VOC]}\right)$ and* k…".

*P12L30 OH + aromatics. To what extent do these reactions involve reversible addition of OH to the aromatic ring, which can then react with O2 (or not in this case as the experiments are in N2). Can the lack of O2 alter the effective (forward) rate coefficient ?*

The lifetime of the OH-aromatic adducts formed by addition of OH to the aromatic ring is approximately 0.3 s at 298 K (Atkinson and Arey, 2003). This lifetime decreases rapidly with increasing temperature. Whilst our experiments are conducted in $N_2$, we estimate that $O_2$ is present in the reactor on the order of $10^{16}$ molecules $cm^3$ due to the use of ambient air to prepare Mixture 3 and impurities in the $N_2$ gas. The reaction between OH-aromatic adducts and $O_2$ is expected to proceed at between 2-8 × $10^{-16}$ $cm^3$ $molecule^{-1}$ $s^{-1}$ for the OH-benzene, OH-toluene and OH-xylene adducts (Atkinson and Arey, 2003). The lifetime of the OH-aromatic adducts, with respect to $O_2$, is therefore likely to be much less than 1 s. For this reason, we anticipate that most of the OH-aromatic adducts do not undergo thermal decomposition to the original aromatic species and OH, and instead react with $O_2$. This is supported by rudimentary kinetic models*, where < 10% of the OH-aromatic adducts undergo thermal decomposition.

This, and other issues concerning OH + aromatic reactions, will be discussed in a future publication focusing on OH + aromatic relative rate measurements.

* Kinetic simulations performed in Kintecus incorporating all aromatic species in Mixture 3. The model included reactions between aromatics and OH forming an OH-aromatic adduct product. This product could then react with $O_2$ with an estimated rate coefficient of 6 × $10^{-16}$ $cm^3$ $molecule^{-1}$ $s^{-1}$ or decay back to the original aromatic species and OH, with an estimated rate coefficient of 2 $s^{-1}$.

*P13L3 "and can only include compounds for which authentic standards or pure raw materials are available." Why pure ? An impure sample will work perfectly well as long as it can be selectively detected. This is the main advantage of relative rate versus absolute methods.*

The purity of the compound is not necessary here: the word "pure" has been removed from the manuscript.

*P13L14 "There is also likely to be a large variety in the concentrations at which atmospheric VOCs exist". I think this is proven rather than "likely".*

Amended to "*Atmospheric VOCs exist at a large range of concentrations…*".

*P13L15 "which could have an adverse effect on the uniformity of individual VOC exposure to OH" What is the "adverse effect" and what does "uniformity of exposure to OH" mean ?*

This sentence has been removed from the manuscript. Whilst the range in VOC concentrations in the atmosphere does represent a practical challenge in real air sampling, it is by no means the most significant.

*P14L15 "OH. However, a significant section of the plot lacks literature data for comparison" In this case it would be beneficial to spike the air with a few reference gases with well-known rate coefficients before entering the reactor.*

The authors agree that, in hindsight, spiking the air with reference compounds would have been beneficial, had we known there would be a considerable gap. However, three of the four monoterpenes would have fit into the aforementioned gap, had they not been depleted to below the limit of detection. A sentence has been added to the manuscript suggesting this: "*It may therefore be beneficial, in future work deriving rate coefficients from real air samples, to synthetically spike the air with a range of reference compounds whose rate coefficients are accurately known, in order to ensure that the best possible correlation between depletion and literature rate coefficient can be generated across the desired range.*"

*P15L13 "this method can produce novel results. . ." perhaps "can produce rate coefficients for VOCs, the OH-kinetics of which have not been investigated to date" or similar.*

Amended to "*We have shown this method can produce rate coefficients for VOCs, the OH-kinetics of which have not been investigated to date…*".

*Figures*

*Figures 3-5 and 7 should be improved by 1) adding the error bars representing the uncertainty in k. I consider this to be very important in order to assess the quality of the data. 2) identifying the VOCs.*

Identifying the VOCs on the plots would improve readers' understanding and interpretation. Figures 3-5 and 7 have therefore been amended, with the data points numbered with their corresponding identity included as part of the figure captions.

Whilst including x-axis (*k* value) uncertainties does aid in comprehension of the linear fit, the authors believe that some of the resultant plots are exceptionally cluttered (see example below). For this reason, plots including the x-axis uncertainties for Mixture 2 will be included as part of the Supplementary Information, whilst the cleaner plot for Mixture 2 remains as part of the main manuscript.

[Figure]

*Figure 7: The inset obscures a large part of the x-axis of the main plot. This can be done better*

*The authors agree that the inset is not necessary and it has therefore been removed.*

**Anonymous Referee #3**

*This manuscript describes flow tube studies in which OH radicals are reacted with a mixture of volatile organic compounds (N>10) simultaneously, in order to generate a self-consistent set of reaction rate coefficients. The concept for the experiment is good, but I question the generality, and hence utility, of the method.*

*The experiments are in general well-described. The manuscript contains clear figures, and also details about dimensions and flow rates used, which always help in doing quick calculations of residence times, etc. The manuscript itself contains some errors and inconsistencies, which will be detailed below. Of course, the simultaneous measurement of rate coefficients by the relative rate method is not new. Kramp and Paulson [J. Phys. Chem. A, 102, 1998, 2685-2690] reacted mixtures of hydrocarbons in Teflon bags to evaluate systematic uncertainties between reference compounds. However, to the best of my knowledge, this has not been done in a flow reactor until now.*

*Kramp and Paulson (1998) used mixtures of three compounds to measure two reference reactions relative to a single target reaction. The target k value was calculated using the reference to target ratios individually, as done in the traditional relative rate method. The method presented in this work differs in that it uses multiple reference reactions to generate a multi-variate relationship, which is then used to derive new k values relative to all other reactions.*

*The biggest issue with the manuscript is that the method is not actually universal, since the derived rate coefficient is dependent on the reactivity of the mixture and the extent of conversion. Thus, it appears that knowledge of the rate coefficients is almost required in order to put them on an absolute basis.*

*Rate coefficients do not depend on reactivity, rather reactivity depends on rate coefficients (see Eq. (4) in the manuscript). Nor do rate coefficients depend on the extent of the conversion of the reactants. However, the rate of a reaction does depend on the extent of conversion; as the reactants are reduced in concentration, the number of collisions reduces. Therefore, the reactor does not suffer the issues suggested by the referee, and has a wide utility for VOCs of differing reaction rates with OH.*

*For this reason, I find that the methodology would have limited applicability to other systems. For example (and this is touched on in the manuscript) in ambient air samples the reactivity can vary widely, and so one needs to know whether the response will be linear or not. Also, if someone else were going to set up a similar experiment, would they have to characterize their system with curves such as those shown in Figure 4? The manuscript does not really answer these questions and reads a little like a work in progress for me, at least.*

*The reactivity of ambient air samples can vary greatly. However, knowing the response (in terms of depletion factor to rate coefficient) is not necessary prior to experiment. Therefore, new reactors could be developed by other groups and applied to novel systems. The key here is that plotting a consistent function through the data allows for the extraction of relative rate data, regardless of whether it is a linear or curved function. A linear function is desirable, as this makes data analysis much easier, but not necessary.*

*For these reasons I can not recommend publication of the manuscript in its current form. Minor corrections follow (format Page/Line).*

*We feel that the explanations provided above counter the referee's concerns. The reactor has wide applicability and can be used without prior knowledge of the shape of the depletion factor versus k response.*

*Page 4/14. The subject of this sentence is "this parameter" (i.e. the reactivity) so it should not vary from milliseconds to tens of seconds.*

The manuscript has been amended to read "*This parameter is equivalent to the inverse of the lifetime of the OH radical which can vary greatly depending on total VOC loading, from milliseconds in heavily polluted areas, to tens of seconds in clean air*" to clarify that the subject should be the OH lifetime.

*Page 4/18. I would imagine that lack of knowledge of the identity of VOCs is more responsible for missing reactivity than errors in the rate coefficients?*

The authors agree that the errors in *k* values are not the issue with the 'missing reactivity', although the sentence in question does not specifically mention errors themselves being the problem. OH reactivity is calculated by combining the concentration of a particular VOC with its rate coefficient for reaction with OH. Hence, the composition of the atmosphere must be known but the identity of each VOC is not strictly needed, although it does help for targeting the measurement of the OH + VOC *k* value.

The manuscript has been clarified to read: "*It is thought that a lack of detailed compositional information and corresponding kinetic data for many OH + VOC reactions may contribute to the so-called "missing reactivity" that has been observed both locally and globally (Lidster et al., 2014).*"

*Experimental section. Sometimes cc are used, sometimes ml for volume. Please be consistent.*

This has been amended for consistency and the manuscript now contains exclusively '$cm^3$'.

*Page 8/3. Remove "a" before "magnitude".*

Amended.

*Page 8/25, and caption for Figure 2. It says here and in the caption that black is lamp off, blue is lamp on, while in the following sentence and the Figure legend this is reversed (and presumably correct).*

Amended.

*Page 9/12, and caption for Figure 3. OH reactivity is given as 180 in text, 240 in Figure caption.*

Amended. It should have been 240 $s^{-1}$ and the text has been clarified to reflect this.

*Page 12, paragraph starting on line 19. This is a little confusing. By non-Arrhenius, I presume you mean having a negative activation energy. This would be expected for addition reactions. However, the last sentence does not really make sense. I do not know if the differences you see are meaningful at this level of precision.*

The sentence has been amended to "*This reflects the temperature dependent relationship described and observed in the literature for OH addition to alkenes and monoterpenes (Chuong et al., 2002; Kim et al., 2011).*"

It is difficult to extract meaningful differences between the measured values for *k* at 298 K and at 323 K. See below for more discussion.

*Page 12/27. I would think that all OH + alkane rate coefficients would show a "normal" activation energy, linear, cyclic or branched.*

The sentences attempting to derive meaningful kinetic information from the *k* values measured at 298 K and 323 K have been removed. The authors agree that it is difficult to extract meaningful differences between the measured values for *k* at 298 K and at 323 K, with the current level of precision.

The section now focuses on the fact that for some of these compounds no 298 K data currently exists and hence, this 323 K data represents our best estimate for room temperature reaction rates.

The section now reads: *"Rate coefficients for the reaction between OH and 12 aromatic VOC at 323 K are also estimated for the first time. We are also able to derive 323 K rate coefficients for the reaction between OH and three alkanes (2-methylheptane, 3-methylheptane and ethylcyclohexane) for which we could find no 298 K data in the literature. This highlights the pressing need for relevant temperature dependent rate coefficient data."*

*Fig 6 Noshino should be Nishino*

Amended.

*McGillen (2008) is missing from ref list*

Amended.